# Global fungal-host interactome mapping identifies host targets of candidalysin

Tian-Yi Zhang[1,6], Yao-Qi Chen[1,6], Jing-Cong Tan[1,6], Jin-An Zhou[1], Wan-Ning Chen[2], Tong Jiang[3], Jin-Yin Zha[4], Xiang-Kang Zeng[5], Bo-Wen Li[1], Lu-Qi Wei[1], Yun Zou[3], Lu-Yao Zhang[3], Yue-Mei Hong[1], Xiu-Li Wang[1], Run-Ze Zhu[1], Wan-Xing Xu[1], Jing Xi[1], Qin-Qin Wang[1], Lei Pan[5], Jian Zhang[4], Yang Luan[1], Rui-Xin Zhu[2], Hui Wang[1]✉, Changbin Chen[3]✉ & Ning-Ning Liu[1]✉

Candidalysin, a cytolytic peptide toxin secreted by the human fungal pathogen *Candida albicans*, is critical for fungal pathogenesis. Yet, its intracellular targets have not been extensively mapped. Here, we performed a high-throughput enhanced yeast two-hybrid (HT-eY2H) screen to map the interactome of all eight Ece1 peptides with their direct human protein targets and identified a list of potential interacting proteins, some of which were shared between the peptides. CCNH, a regulatory subunit of the CDK-activating kinase (CAK) complex involved in DNA damage repair, was identified as one of the host targets of candidalysin. Mechanistic studies revealed that candidalysin triggers a significantly increased double-strand DNA breaks (DSBs), as evidenced by the formation of γ-H2AX foci and colocalization of CCNH and γ-H2AX. Importantly, candidalysin binds directly to CCNH to activate CAK to inhibit DNA damage repair pathway. Loss of CCNH alleviates DSBs formation under candidalysin treatment. Depletion of candidalysin-encoding gene fails to induce DSBs and stimulates CCNH upregulation in a murine model of oropharyngeal candidiasis. Collectively, our study reveals that a secreted fungal toxin acts to hijack the canonical DNA damage repair pathway by targeting CCNH and to promote fungal infection.

Fungal infection poses a great threat to human health. Over one-quarter of the global population suffers from cutaneous (skin) fungal infection and vulvovaginal candidiasis[1]. Furthermore, the mortality rate of systemic fungal infection has reached up to 50% in patients with immunosuppression or immunodeficiency, such as HIV infection, neutropenia, cancer patients[2–4].

The pathogen-host interactions (PHI), particularly protein-protein interactions (PPIs), underlie the process of infection[5]. Fungal pathogens are capable of adhering to, colonizing, and invading broad host niches, causing superficial to life-threatening invasive fungal infections[6]. To protect the host against fungal infection, inherent defense systems such as the protective immune response exist[7].

[1]State Key Laboratory of Systems Medicine for Cancer, Center for Single-Cell Omics, School of Public Health, Shanghai Jiao Tong University School of Medicine, Shanghai 200025, China. [2]Department of Gastroenterology, The Shanghai Tenth People's Hospital, Department of Bioinformatics, School of Life Sciences and Technology, Tongji University, Shanghai, China. [3]The Center for Microbes, Development, and Health, Key Laboratory of Molecular Virology and Immunology, Unit of Pathogenic Fungal Infection & Host Immunity, Shanghai Institute of Immunity and Infection, Chinese Academy of Sciences, Shanghai 200031, China. [4]State Key Laboratory of Systems Medicine for Cancer, Key Laboratory of Cell Differentiation and Apoptosis of Chinese Ministry of Education, Shanghai Jiao Tong University, School of Medicine, Shanghai 200025, China. [5]The Center for Microbes, Development, and Health, Key Laboratory of Molecular Virology and Immunology, Shanghai Institute of Immunity and Infection, Chinese Academy of Science, Shanghai, China. [6]These authors contributed equally: Tian-Yi Zhang, Yao-Qi Chen, Jing-Cong Tan. ✉e-mail: huiwang@shsmu.edu.cn; cbchen@siii.cas.cn; liuningning@shsmu.edu.cn

Similarly, viral and bacterial pathogens also manipulate host cellular mechanisms that facilitate pathogen proliferation and evasion of immune response through PPIs[5,8]. For example, *Salmonella enterica* serovar Typhimurium (STm) can secrete effector proteins to hijack host cellular processes, leading to severe host damage[3,7,9]. Thus, investigation of the architecture and dynamics of fungi-host interaction may unveil targets against fungal infection.

The fungal microbiome (mycobiome) inhabit in diverse human body sites and play a critical role in the induction and function of host immune system[10–12]. *Candida albicans* is the leading cause of healthcare-associated bloodstream infections in U.S. It can transit between commensalism and pathogenicity by interacting with host in a complicated and dynamic way[10,13,14]. While it normally colonizes the oral, gastrointestinal, or genital tracts as the commensal, it will transform into a pathogen in immunocompromised or microbiome dysbiosis conditions, leading to systemic or mucosal candidiasis[7,15–17]. The

yeast-to-hyphae morphological transition is critical for *C. albicans* virulence[16,18]. *ECE1* was one of the genes to be identified in hyphal-specific expression, yet until now it has been one of the most poorly understood genes in *C. albicans*. Moyes et al. discovered that Ece1 is secreted from hyphae as a group of eight short peptides (Ece1-I to Ece1-VIII) and demonstrated that Ece1-III (candidalysin) mediates the pathogenic activity associated with *ECE1*[18]. Candidalysin, encoded by *ECE1*, was a hypha-associated, α-helix amphipathic, 31-amino-acid, and non-proteinase peptide toxin. It triggers a series of cellular stress responses that induce necrotic death, including reactive oxygen species (ROS) production, intracellular ATP depletion, mitochondrial dysfunction, and cytochrome C release[18,19]. As a potent trigger of alarmin and antimicrobial peptide release in epithelial cells[20], candidalysin can induce epithelial damage and activate the innate epithelial immune response via an epidermal growth factor receptor (EGFR)[15,21]. Particularly, candidalysin triggers innate immunity during fungal

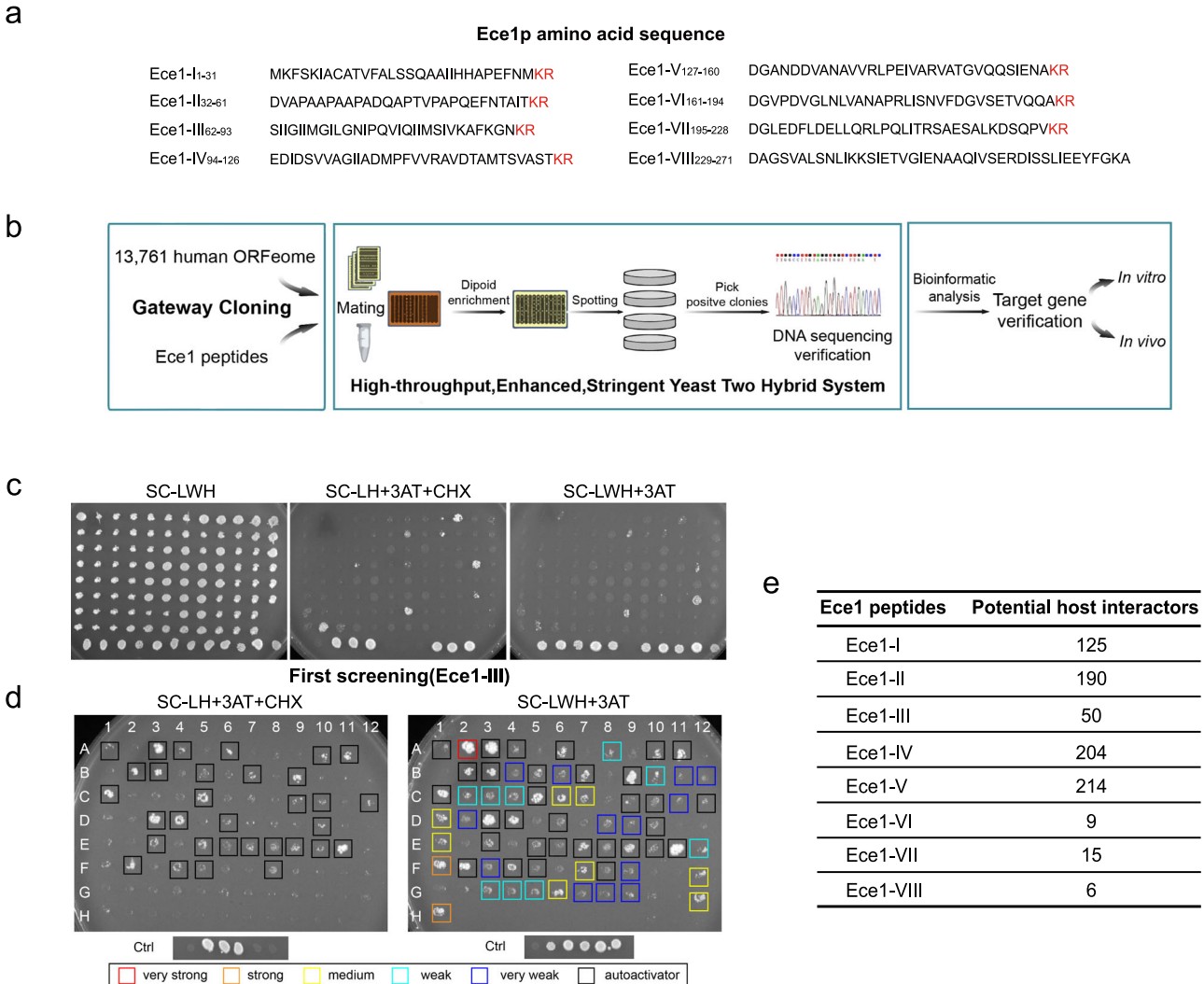

**Fig. 1 | HT-eY2H screening of potential human interactors for Ece1-derived peptides. a** The amino acid sequence of each Ece1 peptide. **b** The schematic workflow based on the high-throughput enhanced yeast two-hybrid (HT-eY2H) system to conduct interactome analysis between each Ece1 peptide ($n = 8$) and genes in the human ORFeome ($n = 13,761$). **c** The plate dotting images of the initial HT-eY2H screening on three types of selective media involving exposure to the candidalysin peptide (Ece1-III). **d** The plate dotting images of the 2nd HT-eY2H screening on two types of selective media involving exposure to the candidalysin peptide (Ece1-III). Columns 1–6 of Controls: 1 expresses DB and AD plasmids

without any fusion; 2 expresses DB-pRB and AD-E2F1 fusion proteins, forming an interaction, and is CHX-sensitive; 3, 4, and 5 express DB-Fos and AD-Jun, DB-GAL4 and AD, and DB-DP and AD-E2F1, respectively, all of which exhibit positive interactions but are CHX-resistant; 6 expresses DB-DP and AD-E2F1, and is CHX-sensitive. The annotation of 1-12 and A-H facilitates our localization of each protein. The colonies were color-categorized according to growth intensity as very strong (red), strong (orange), medium (yellow), weak (cyan), and very weak (blue). Auto-interactive colonies were marked with black squares. Ctrl, Control. **e** Numbers of potential human protein targets of each peptide.

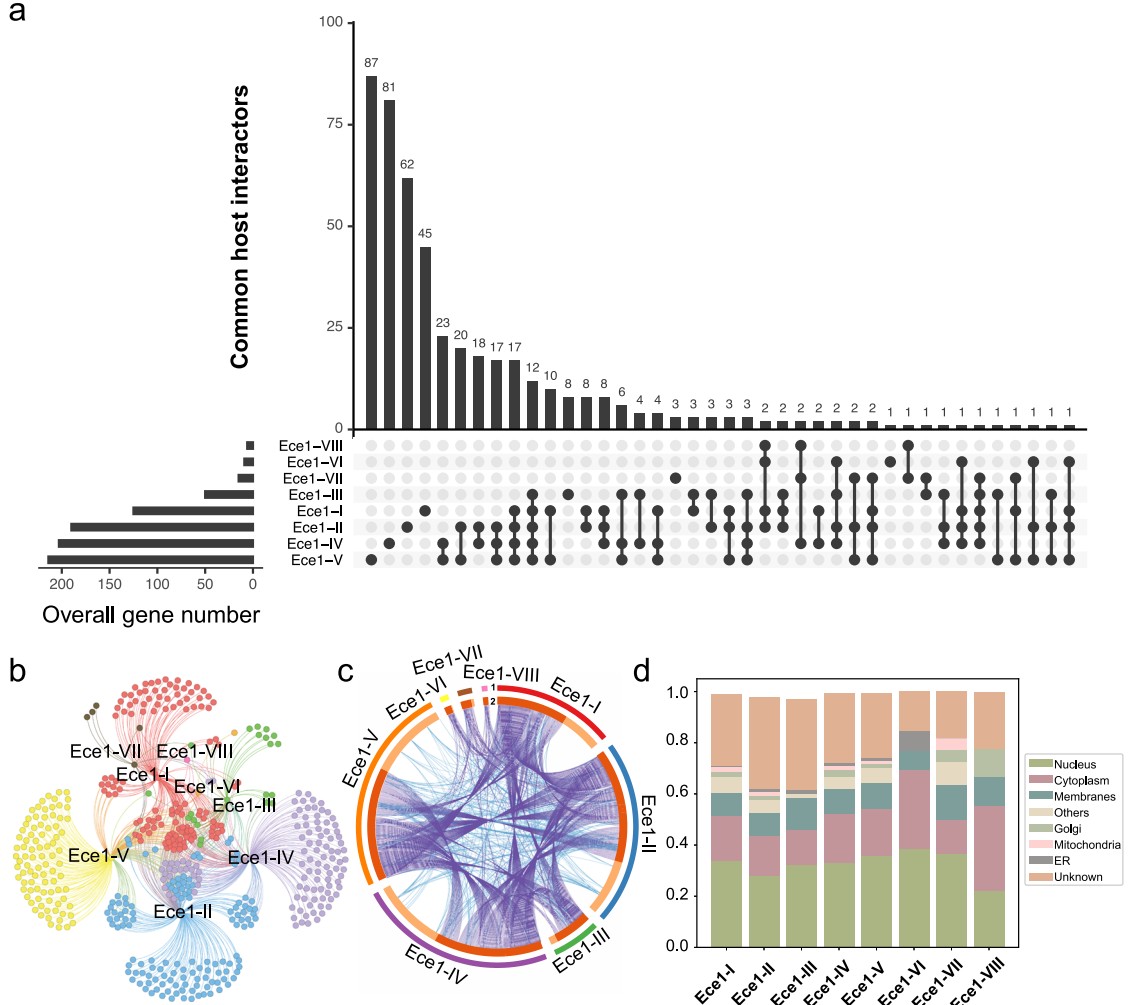

**Fig. 2 | Global interactome between human and Ece1-derived peptides. a** Bar chart showing total human interactors for each of the eight peptides with shared interactors displayed in the histograms as connected dots. **b** Interaction network of the Ece1 peptide-human ORFeome whereby each peptide is assigned with a different color: Ece1-I (red), Ece1-II (blue), Ece1-III (green), Ece1-IV (purple), Ece1-V (yellow), Ece1-VI (orange), Ece1-VII (brown), and Ece1-VIII (pink). Overlapped genes are centrally located in the diagram, and the noted genes were commonly shared among multiple peptides. **c** Circos plot involving the eight Ece1 peptides, whereby the outer ring (color-coded as in panel **c**) represents the genes interacting with each peptide, while the inner ring represents genes shared between two or more peptides (dark orange) or genes solely interacting with the individual peptide (light orange). Purple lines connect the peptides that share the interaction genes and blue lines connect genes belonging to the same functional group. **d** Stacked bar graph representing the relative abundances of subcellular localization involving human interactors with each Ece1 peptide. The 'Others' category includes: Actin filaments, Centrosome, Microtubules, Focal Adhesion Sites, Intermediate Filaments, Peroxisomes, Microtubule Ends, Cell Junctions, Midbody Ring, Endosomes, Lysosomes, Centriolar Satellite, and Cytokinetic Bridge.

infection through proinflammatory mediator release, neutrophil recruitment and Type 17 immunity[22,23]. However, candidalysin has evolved dual functions to counteract macrophage antimicrobial activities and evade immune surveillance. It can exist as a classical virulence factor and an immunoavoidance factor during the *C. albicans*-macrophage interaction, as well as mucosal and systemic infection. It was reported that candidalysin, as a key fungal avirulent factor, promoted antifungal T cell responses in a platelet-expressed GP1ba dependent manner[24] and activated the IL-1β-CXCL1 pathway to instigate protective host central nervous system (CNS) immunity[25]. However, the direct human targets of candidalysin and the related mechanisms in promoting systemic fungal infection have not been fully defined.

In this study, a HT-eY2H assay was used to map the global interactomes of Ece1 peptides, including candidalysin, with a part of the human ORFeome (n = 13,761). Our study uncovered the potential host target of candidalysin and provided the molecular determinant for fungi-host interaction that critically impacts fungal infection. These results not only promote our understanding of the pathogenesis of *C.*

*albicans* infection, but also provide therapeutic insights against fungal infection.

## Results

### Identification of host interactors with Ece1 peptides by HT-eY2H screening of the human ORFeome library

Although Ece1 can be secreted from hyphae as a group of eight short peptides[18] (Fig. 1a), only Ece1-II, -III, -V, and -VI were identified in culture supernatants by LC-MS. However, it is still possible that the other undetected peptides may also exert biological functions during *C. albicans* infection. We thus first examined whether peptides like Ece1-I, -IV, -VII, and -VIII could have impacts on the gene expression of human cells by performing the transcriptome analysis. Our results demonstrated that treatment with each of the four Ece1 peptides significantly influenced the global transcription and biological functions of FaDu cells in vitro (Supplementary Fig. 1, Supplementary Data 1 and 2), highlighting the potential importance of all these secreted peptides during the host-*C. albicans* interactions. To identify the host interactors of candidalysin, we performed a high-throughput enhanced

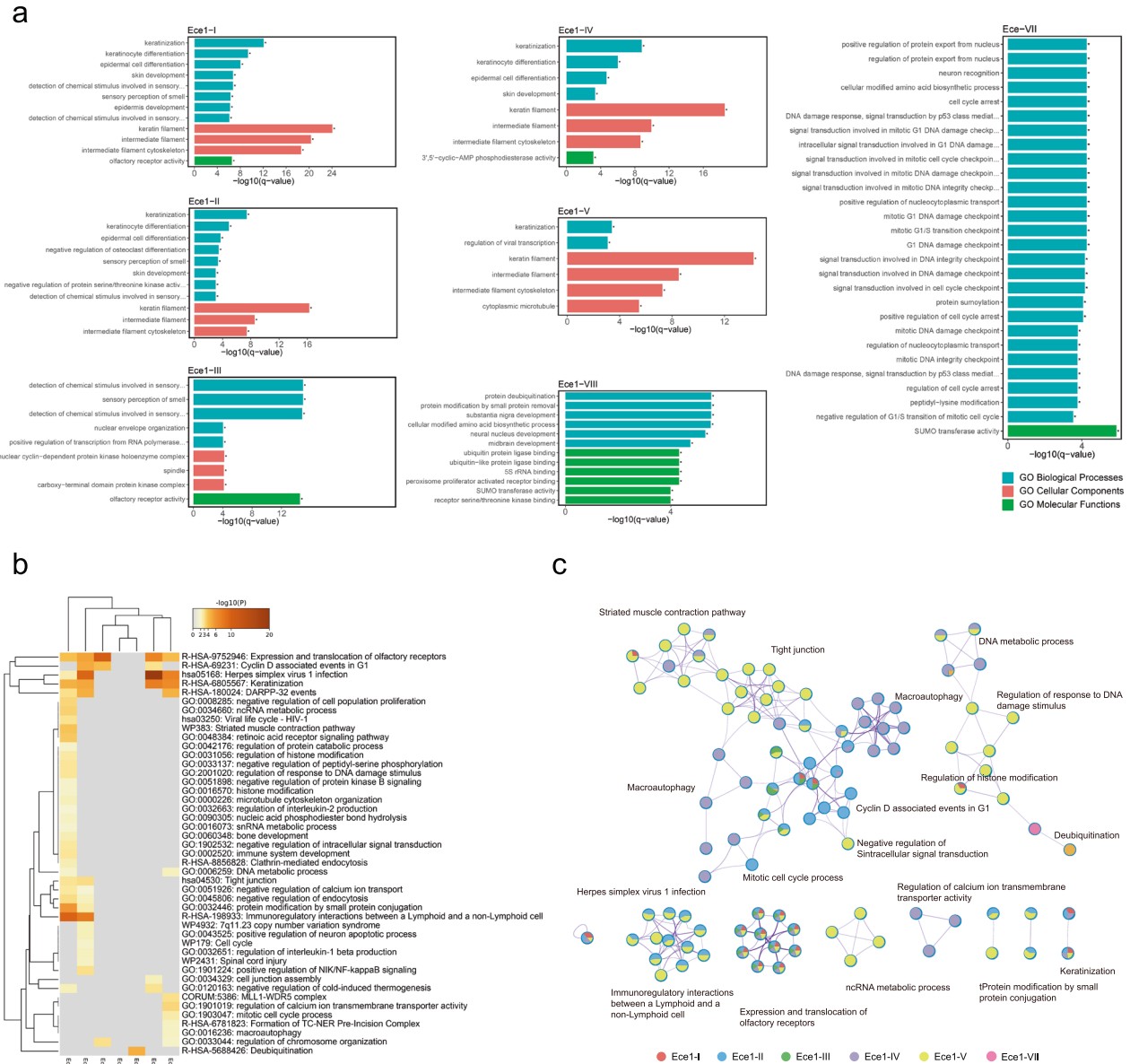

**Fig. 3 | Functional analysis of interactions between each Ece1 peptide and the human ORFeome. a** Histograms of GO enrichment analysis associated with each Ece1 peptide. The top significantly enriched features are shown by categories of Biological Processes, Cellular Components, and Molecular Functions. *q*-values are calculated using the Benjamini-Hochberg procedure. **b** Heatmap of feature enrichment analysis for the eight Ece1 peptides, for which the first 100 features (right side) were selected to plot the heatmap. It is colored by *P*-value, where darker colors indicate lower values (i.e., greater enrichment) and gray color indicates the peptide is not enriched for the associated feature. *P*-values were calculated based on the cumulative hypergeometric distribution. **c** Networks of GO enrichment analysis were inferred for features associated with each color-coded Ece1 peptide: I (red), II (blue), III (green), IV (purple), V (yellow), VI (missing), VII (pink), and VIII (missing). Each circle represents all the genes in the same pathway categorized by different peptides, and circles containing multiple colors indicate multiple peptides share the feature.

yeast two-hybrid (HT-eY2H) screening of the human ORFeome (Fig. 1b), which we used previously[26]. The Ece1 peptides and the human ORFs were both transformed and expressed in yeast cells and plated on three types of selective media (Fig. 1b). These plates were compared to confirm the positive activation of GAL1:HIS3 in a CHX-sensitive manner as well as the elimination of DB-X auto-activators. The identities of the positive clones were confirmed by Sanger sequencing. A global interactome map was then inferred to explore direct binary protein-protein interactions (PPIs). The screening results on selective media were shown in both Fig. 1c, d. In addition to identifying what the interactors were, we also probed the strength of interaction as either very weak, weak, medium, strong, and very strong (Fig. 1d). Overall,

our global interactome map provided candidalysin-specific interactors and additional host proteins unexpectedly interacting with the other seven Ece1 peptides. A total of 813 high-confidence interactions inferred among 13,761 potential human protein targets and each of the eight Ece1 peptides. The numbers of potential interactors (Fig. 1e) varied across the eight different Ece1 peptides, ranging from as few as 6 (Ece1-VIII) to as many as 214 (Ece1-V) (Supplementary Data 3). In particular we discovered 50 potential host interactors of candidalysin (Ece1-III). We also uncovered potential interactors for the other seven Ece1 peptides previously reported to be unable to damage epithelial cells, or to induce cytokine secretion or p-MKP1/c-Fos-mediated danger responses[18].

## A global Ece1 peptides–host interactome map

To compare the interactome and identify specific host interactors for each Ece1 peptide, we first examined the overall host interactor numbers across peptides. Although they were varied greatly with each peptide, the overlapping factors were discovered (Fig. 2a). For example, twelve genes consistently interacted with Ece1-I, Ece1-II, Ece1-III, Ece1-IV, and Ece1-V, including *PPP2CA*, *OR7C2*, and *OR7A10*. In addition, we found that both *RHOA* and *UBTD2* interacted with Ece1-I, Ece1-II, Ece1-VI, Ece1-VIII, while *CCNH*, *GLRX3*, and *UBXN2B* interacted with Ece1-II, Ece1-III, Ece1-IV, and Ece1-V. Notably, candidalysin had the highest overlapping interactors with Ece1-I, Ece1-II, Ece1-IV, and Ece1-V (Fig. 2a and Supplementary Data 4) while Ece1-VIII had the least.

A significant number of genes that interacted with multiple peptides were also identified (Supplementary Data 4). For example, *ACD* interacted with six Ece1-peptides except for Ece1-VI and Ece1-VIII. *TADA2B* also interacted with six Ece1-peptides, excluding Ece1-III and Ece1-IV. After removing the overlapped host interactors, a list of Ece1 peptide specific host factors were identified. Except for Ece1-VIII, we observed specific interactors for all the other seven peptides, including 45 genes for Ece1-I, 62 for Ece1-II, 8 for Ece1-III, 81 for Ece1-IV, 87 for Ece1-V, 1 for Ece1-VI, and 3 for Ece1-VII. In addition, we found unexpected human interactors for candidalysin such as *IST1*, *CBWD6*, and *CBWD2*. *IST1*, required for the formation of endosomal sorting complex III (ESCRT-III)[27], has been associated with microtubule-interacting and transport (MIT) domain binding in GO enrichment analysis. When comparing the correlation of overlapping factors between two different peptides, we found that candidalysin achieved significantly more overlaps in its fungus–host protein–protein interactions with Ece1-II, Ece1-V, and Ece1-VII. Ece1-II, the adjacent genomic neighbor to Ece1-III, was observed with the most significant overlaps among all the other peptides, which was reported likely to form an indispensable part in Ece1 polypeptide cleavage[28]. The relatively large number of shared genes between these peptides may be a result of the similar molecular interaction patterns.

We next examined the interactome networks of all Ece1 peptides. Specific clusters for each Ece1 peptide were identified (Fig. 2b). Several hub-node factors were uncovered in the network, such as *CCNH* (interacting with Ece1-II, III, IV, and V), *RHOA* (interacting with Ece1-I, II, VI, and VIII), *PPP2CA*, (interacting with Ece1-I, II, III, IV, and V), etc. Obviously, the numbers of shared interactors for each peptide were more than those of unique genes, indicating potential interplay among these Ece1 peptides (Fig. 2c). Regarding to subcellular localization, more host interactors existed in nucleus than that in cytoplasm (Fig. 2d). For example, most of the Ece1-VI-interacting genes located in nucleus, while most of Ece1-VII interactors existed in cell membrane, suggesting potential cell entry of *C. albicans* through peptide interactions with these human proteins in addition to known mechanisms. Altogether, our results uncovered Ece1-peptide specific and shared protein-protein interactomes, suggesting preferential selection for hub genes that may be crucial for the biological roles of Ece1 peptides.

## Functional enrichment of host interactors specific to each Ece1 peptide

The Ece1 peptide-specific host interactors necessitated the exploration of functional pathways that were significantly represented in each interactome. Gene Ontology (GO) analysis was performed to cluster host interactors into different functional modules, including a statistical enrichment histogram of specific biological processes, cellular components, and molecular functions (Fig. 3a and Supplementary Fig. 2). We found that a large proportion of host interactors with candidalysin (Ece1-III) were related to sensory perception of chemical stimuli and olfactory receptor activity, with most interactors in the spindle components. Unexpectedly, the top-ranking biological process for each of Ece1 peptides I, II, IV, and V was keratinization. The enriched biological processes of Ece1-VII mostly related to protein

export from the nucleus and DNA damage repair, while those for Ece1 peptides VI and VIII were mainly involved in organelle organization and protein deubiquitination, respectively.

A comparative analysis across all eight Ece1 peptides was then presented as a functional enrichment heatmap (Fig. 3b). It was indicated that the olfactory receptor activity pathway was ranked first to be closely related with candidalysin. Interestingly, both fungal infection and mycotoxin production have been reported to be inextricably linked with sensory neural pathways[29]. Indeed, we found that mice treated with candidalysin spent more time trying to seek food (longer latency to retrieve food), suggesting potential detrimental effects of candidalysin on host olfactory function. Compared to those of the wild type (WT) *C. albicans*, both the olfactory index and Anxiety Avoidance (AA) index were decreased in fruit flies challenged by the candidalysin deletion mutant, suggesting that the AA avoidance behavior was compromised in *C. albicans* following the loss of candidalysin (Supplementary Fig. 3a and b). However, we cannot exclude the possibility that the detrimental effects of candidalysin treatment on host olfactory function might also be due to the destruction of olfactory cells, considering the property of a cytolysin.

Moreover, we observed significantly enriched pathways related to cell cycle and cell proliferation, cytoskeleton and cell connection, infection and immune regulation (Fig. 3b). Ece1-V was also observed to significantly interact with host factors such as regulators of DNA damage response, histone modification, DNA and ncRNA metabolic pathways (Fig. 3c). These results were consistent with the fact that human mycobiome, especially the *Candida* genus, reported to be critical for tumor progression[30–34], such as *C. albicans* with oral cancer through induction of DNA damage, epithelial membrane damage, and overexpression of inflammatory signaling pathways[35]. In addition, the functional enrichment network showed that the shared host interactors of both the candidalysin (Ece1-III) and Ece1-II peptides were associated with the Cyclin D associated events in G1 pathway regulated by amounts of interactors, among which candidalysin interactors are as follows: *ACD*, *PPP2CA*, *UBXN2B*, *CCNH*, *CDX1*, *RB1*, *SFI1*, *SPRED2*, *THPO*, *SPAG8*, *NEK6*, *IST1* (Fig. 3c). These results provide functional insights into each Ece1 peptide.

## Candidalysin-induced DNA damage through upregulation of γ-H2AX formation and CCNH expression

To examine the damage induced by candidalysin, we treated both FaDu and A549 epithelial cells with synthetic candidalysin peptide. It induced both cell damage and cytokine secretion (Supplementary Fig. 4a). LDH release was significantly increased with incremental Ece1-III dosages range from 5 to 80 μM (Supplementary Fig. 4b). The prolonged incubation with 10 μM Ece1-III for 12 h induced the secretion of IL-6 and G-CSF (Supplementary Fig. 4c, d). Among the candidalysin interactors mentioned above in our Y2H screening, CCNH, known to mainly regulate DNA metabolism and damage repair, cell cycle related pathways, etc.[36,37], was found to interact with Ece1-II to V peptides by both co-immunoprecipitation (Co-IP) and BIAcore assays (Supplementary Fig. 5a, g, h). Therefore, we first analyzed cell cycle in the CHO cells by flow cytometry after treatment with 10 μM candidalysin, a concentration that was proved to be non-toxic (Fig. 4a). As expected, 10 μM candidalysin treatment effectively blocked cell cycle in the G2/M phase, while a higher dose (20 μM) blocked cells in the S phase (Fig. 4b).

CCNH, the highly conserved cyclin, can function as the cyclin-dependent kinase (CDK)-activating kinase (CAK) complex[38], affecting both DNA damage and DNA repair signaling pathways[39]. We next assessed whether candidalysin could induce DNA damage. After exposure to 10 μM candidalysin for 4 h, the incidence of double-strand DNA breaks (DSBs) in the candidalysin-treated group was significantly increased as shown by greater DNA migration distance (Fig. 4c). Accordingly, we observed a higher incidence of micronucleus

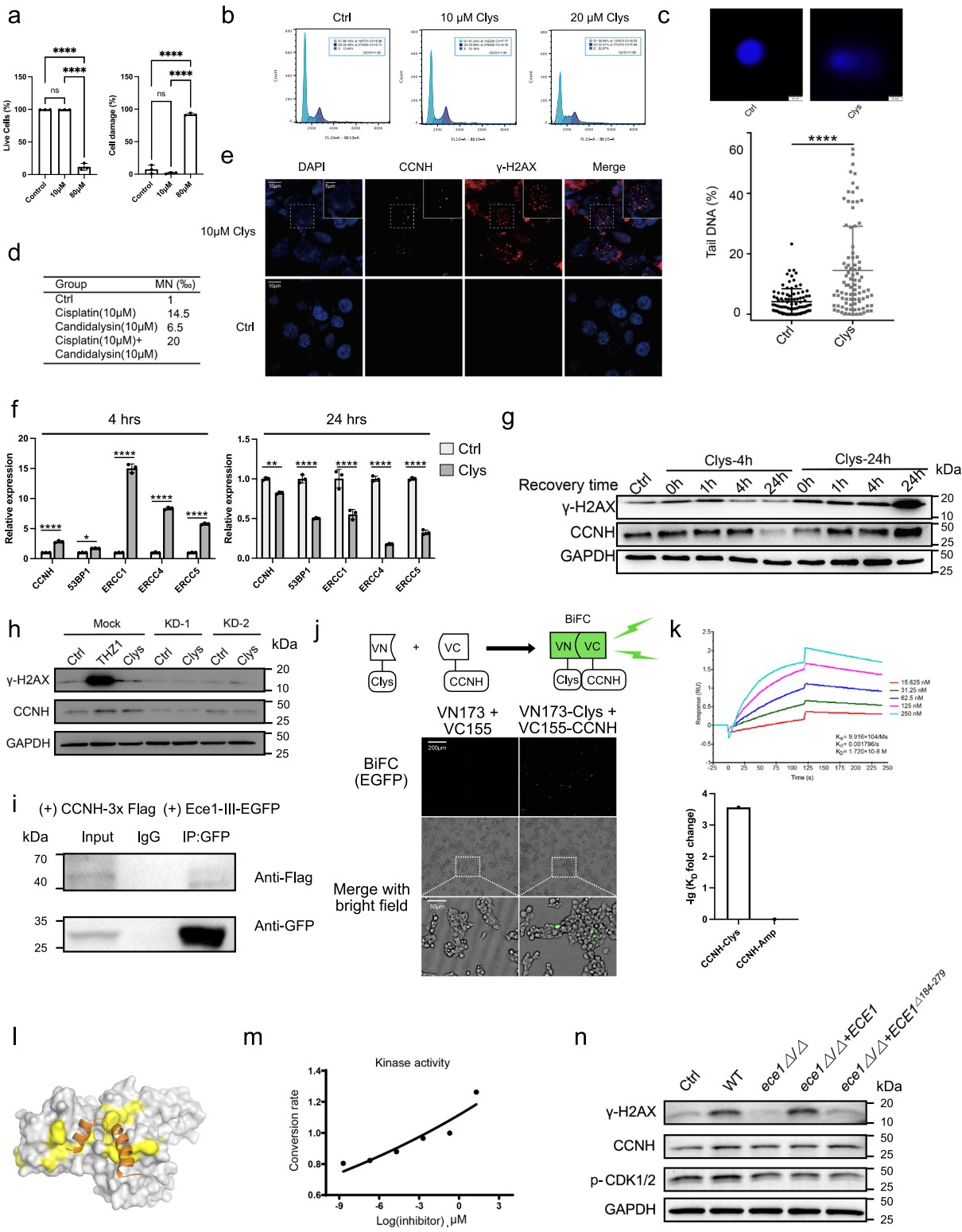

formation in candidalysin treated FaDu cells (Fig. 4d). In addition, we also discovered that *RHOA*, which interacts with Ece1-I, II, VI, VIII in our screen, had the potential to be a host interactor as shown by increased mRNA and protein expression (Supplementary Fig. 5b). RHOA, a member of the Rho family of small GTPases[40], could promote the reorganization of actin cytoskeleton and was associated with tumor cell proliferation and metastasis[41].

The H2AX phosphorylation, a key step in DNA damage repair (DDR), could increase DNA accessibility and recruit specific DDR proteins at DNA ends through chromatin modification. We thus examined one of the most deleterious types of DNA damage, DSBs[42,43], by immunofluorescent detection of γ-H2AX formation, which is a reliable biomarker for DSBs rejoining[44]. We found that candidalysin triggered the accumulation of γ-H2AX, which was colocalized with CCNH and recruited at DNA damage sites (Fig. 4e). However, unlike candidalysin, the non-toxin cytolysin like low-dose Triton X-100 or some other pore forming toxins such as mellitin or hemolysin all failed to induce severe DNA damage (Supplementary Fig. 5d).

**Fig. 4 | Candidalysin induces double-strand DNA breaks and suppresses DNA damage repair by binding directly to CCNH. a** Viability and death rate of FaDu cells in control, 10 μM and 20 μM candidalysin-treated group (4 h) ($n = 3$ samples), detected via Calcein/PI viability/cytotoxicity assay. Data was shown as mean ± SD. The $P$ value was determined by one-way ANOVA with Tukey's post-hoc tests. ****$P < 0.0001$. **b** Cell-cycle analysis of CHO-K1 cells after 10 μM or 20 μM candidalysin treatment for 4 h. G2/G1 indicates the ratio of DNA fluorescence intensity in G2/M phase cells to DNA fluorescence intensity in G1 phase cells. Example of the illustration: the proportion of G1 phase cells in the Control group was 64.14%, the average fluorescence intensity was 140721, and the CV was 6.86; the ratio of G2 phase cells to G1 phase cells was 1.95. The $P$ value was determined by two-sided Pearson's Chi-Square test, $P < 0.001$. **c** The representative image of comets enriched in FaDu cells. The cells were stimulated by 10 μM of candidalysin for 4 h. Percentage of tail DNA (%) in control and candidalysin-treated group ($n = 90$ cells). Ctrl, Control; Clys, candidalysin. Data was shown as the median with interquartile range. The $P$ value was determined by two-sided Mann–Whitney $U$ test. $P < 0.0001$. **d** The micronucleus (MN) ratio induced by candidalysin in FaDu cells. **e** The γ-H2AX formation and CCNH expression in FaDu cells. The representative image (from 3 biological repeats) indicates the γ-H2AX formation (red) and CCNH expression (green) in the nuclei. The expression of CCNH and γ-H2AX was evaluated by the immunofluorescence assay. Scale bar, 10 μm, 5 μm, respectively. **f** The transcriptional expression level of DNA damage repair and nucleotide excision repair-related genes in FaDu cells stimulated by candidalysin. Cells were stimulated by 10 μM of candidalysin for 4 h and 24 h. The mRNA expression levels were measured by qRT-PCR ($n = 3$ samples). Data was shown as mean ± SD. The $P$ value was determined using two-way ANOVA followed by Šídák's multiple comparisons test.

*$P = 0.0112$; **$P = 0.001$; ****, $P < 0.0001$. **g** The DNA damage repair efficiency in FaDu cells. The FaDu cells were stimulated with 10 μM of candidalysin for 4 h or 24 h, respectively. The cyclin H (CCNH) and γ-H2AX protein expression levels were measured by western blot (from 3 biological repeats). **h** Protein expression levels of CCNH and γ-H2AX in A549 stimulated by 10 μM THZ1 or 10 μM candidalysin, and in CCNH-KD A549 stimulated by 10 μM candidalysin for 4 h (from 3 biological repeats). **i** Co-immunoprecipitation assay demonstrated the interaction between CCNH and candidalysin in HEK-293T cells (from 3 biological repeats). **j** The bimolecular fluorescence complementation (BiFC) assay by transfecting N-terminal EGFP-tagged Clys and C-terminal EGFP-tagged CCNH into HEK-293T cells. The green fluorescence signal confirmed the interaction between Clys and CCNH proteins within the host cells (from 3 biological repeats). **k** Upper panel: the binding efficiency of candidalysin to CCNH protein was evaluated by BIAcore. By curve fitting, we measured the binding rate constant $K_a = 9.916 \times 10^4$/Ms, dissociation rate constant $K_d = 0.001706$/s, and equilibrium dissociation constant $K_D = 1.720 \times 10^{-8}$ M for candidalysin. Lower panel: comparison of equilibrium dissociation constant ($K_D$) of CCNH-Clys binding and CCNH-Amp binding. Amp, human cathelicidin (FKRIVQRIKDFLRNLVPRTES). $K_D$ fold change = $K_D$(CCNH-Clys)/ $K_D$(CCNH-Amp). **l** Docking patterns performed by HPepDock. The structure of candidalysin was predicted by AlphaFold and was colored as orange. **m** The kinase activity of CAK complex induced by candidalysin in vitro. **n** The protein expression levels of CCNH and γ-H2AX and phosphorylation level of CDK1/2 (substrate of CAK complex) were measured by western blot after co-culturing FaDu cells with the *C. albicans* WT, *ece1Δ/Δ, ece1Δ/Δ+ECE1* and *ece1Δ/Δ+ECE1^{Δ184-279}* (from 3 biological repeats). WT, wild type; *ece1Δ/Δ*, null mutant; *ece1Δ/Δ+ECE1*, *ECE1* re-integrant; *ece1Δ/Δ+ECE1^{Δ184-279}*, candidalysin re-integrant.

## Candidalysin inhibits DNA damage repair by binding directly to CCNH

We next tested if the downstream DNA damage repair (DDR) pathway associated genes were affected. The p53-binding protein 1 (*53BP1*) is a well-known DDR factor[45]. The XPF (ERCC4)-ERCC1 heterodimer, along with XPG (ERCC5), are downstream factors in the NER pathway after the CAK subcomplex dissociates from TFIIH[46]. We hypothesized that treatment with candidalysin would affect the efficiency of DDR. By RT-PCR assay, we found the transcriptional expression of *CCNH* and NER-related downstream factors (including *ERCC1, RCC4/XPF* and *ERCC5/XPG*), as well as DDR-related gene, *53BP1*, was upregulated after short-term exposure to candidalysin (4 h). In contrast, long-term exposure for 24 h to candidalysin transcriptionally suppressed these genes (Fig. 4f and Supplementary Fig. 5c). Importantly, a long-term exposure of candidalysin leads to sustained higher expression of γ-H2AX even during the recovery period, an indicator of ineffectiveness of the DDR. For comparison, following the short-term exposure, the expression level of γ-H2AX was gradually restored to control level (Fig. 4g). Intriguingly, we observed a dose-dependent inhibition of CCNH expression by candidalysin after the 24 h treatment, while γ-H2AX level remained elevated. This observation suggests that a long-term exposure of candidalysin may cause irreversible effects on DDR.

To determine whether CCNH was specifically required for the downstream γ-H2AX formation, we knocked down *CCNH* in the A549 epithelial cell line. As shown in Fig. 4h, treatment with either 10 μM THZ1 (a selective and potent covalent CDK7 inhibitor) or candidalysin for 4 h resulted in prominently elevated CCNH and γ-H2AX expression, consistent that found in FaDu cells. However, when *CCNH* was knocked down, the γ-H2AX expression was maintained at low levels in the presence of candidalysin similar to that of control. Thus, the low level of γ-H2AX observed may result from inhibition of the DDR pathway mediated by CCNH in response to candidalysin-induced DNA damage. This notion was further confirmed by antibody blockade of CCNH, as we found that CCNH inhibition significantly suppressed the expression of γ-H2AX (Supplementary Fig. 5f), supporting the role of CCNH in mediating Clys-induced inhibition of DNA damage repair.

As shown in Fig. 4i, the direct interaction between intracellular candidalysin and CCNH was verified by the Co-IP assay. To confirm this interaction, we further performed the bimolecular fluorescence

complementation (BiFC) assay by transfecting N-terminal EGFP-tagged candidalysin and C-terminal EGFP-tagged *CCNH* into HEK-293T cells. Clearly, the formation of green fluorescence signals strongly reflects the intracellular interaction between candidalysin and CCNH (Fig. 4j).

To confirm that candidalysin drives CDK7 activation by CCNH, we purified the CCNH protein and performed Surface Plasmon Resonance (SPR) analysis on a BIAcore T200 instrument. The binding kinetics of candidalysin and CCNH indicated the real-time- and dose-dependent-binding association and dissociation rates with increased concentrations of candidalysin (Fig. 4k). The $K_a$ and $K_d$ values denoted the binding and dissociation velocities of CCNH protein and candidalysin, respectively. The $K_D$ reflects the binding strength, i.e. affinity. A close inspection of the binding profiles shown in Fig. 4k indicated the stronger binding with CCNH at higher concentrations. At lower concentrations (15.625 nM), a response was also discernable on the CCNH surface. At the highest DN280 concentration (250 nM), the overall binding response was greater for CCNH. This was presumably due to the higher capacity of CCNH surface because more CCNH had been immobilized during the streptavidin-capturing step. After curve fitting of the concentrations versus equilibrium response data, the equilibrium dissociation constant for candidalysin reached up to 17.2 nM, demonstrating strong affinity for CCNH. Thus, candidalysin recognition is tightly linked to the presence of CCNH and it can bind directly to CCNH with strong affinity (Supplementary Fig. 5g, h).

To gain a structural insight into this interaction, we further performed the molecular docking pattern of CCNH (PDB ID: 1KXU[47]) and candidalysin by HPepDock[48]. The predicted docking pattern of candidalysin is shown in Fig. 4k. We observed that the binding of candidalysin to CCNH was mostly triggered by hydrophobic interaction, instead of a moderate H-bond between Trp11 on CCNH and Gly11 on candidalysin (Supplementary Fig. 6b), and the top 5 residues contributing the binding is Ile5, Ile3, Ile13, Pro14, and Gly4 (Supplementary Fig. 6b and Supplementary Table 3). Importantly, both MM/GBSA and short time molecular dynamic simulation ensure the interaction stability between CCNH and candidalysin.

We postulated that the binding of candidalysin to CCNH might be critical for activation of CAK activity to regulate DNA damage repair. After measuring ADP production from a kinase reaction with the ADP-

Glo kinase assay, we observed that CAK kinase was activated gradually with increasing concentrations of candidalysin (Fig. 4m). Moreover, the increasing level of phosphorylated CDK1/2 also confirmed the activated CAK kinase (Fig. 4n). As CAK can regulate NER negatively[49], this dose-dependent activation suggested that binding of candidalysin to CCNH activated the CAK kinase to inhibit DDR. The importance of this finding is that candidalysin not only induces DNA damage, it may have evolved to specifically recognize CCNH and activate CAK complex, leading to suppression of DDR.

### Candidalysin-induced DNA damage is critical for mucosal pathogenesis

To test the in vivo efficacy of candidalysin on DNA damage, a murine model of oropharyngeal candidiasis was conducted by challenging with both candidalysin and each of the following *C. albicans* strains: WT, an *ECE1* null mutant (*ece1Δ/Δ*), an *ECE1* re-integrant (*ece1Δ/Δ +ECE1*), and a mutant allele lacking the candidalysin-encoding region of *ECE1* (*ece1Δ/Δ+ECE1^Δ184-279^*) (Fig. 5a). The fungal load was significantly decreased for the *ece1Δ/Δ* and *ece1Δ/Δ+ECE1^Δ184-279^* mutants (Fig. 5b). To assess the in vivo genotoxicity of candidalysin, we evaluated the frequencies of mature and immature micro-nucleated erythrocytes in both peripheral blood and bone marrow cells of mice. The ratio of MN PCE (micro-nucleated polychromatic erythrocytes) in the *ece1Δ/Δ* and *ece1Δ/Δ+ECE1^Δ184-279^* infected mice was significantly reduced compared to either WT or the group treated with candidalysin alone, indicating the clastogenic activity of candidalysin (Fig. 5c). Histological analysis of sections of paraffin-embedded tongue tissue using both H&E and PAS staining revealed extensive hyphal invasion into the keratinocyte mucosal barrier and tissue damage with infiltrated neutrophils in both WT and *ECE1* re-integrant infected mice (Fig. 5d). This was further evidenced by the fact that the tongue tissues exhibited an irregular epithelium, including more micro-differentiation and para-keratinization of the upper layers. Underneath the epithelium, a large population of inflammatory cells from infiltration could be observed among the connective tissue fibers. By contrast, the epithelial layer of tongue tissue was minimally damaged after exposure to the *ECE1* deletion mutants, and *C. albicans* was barely visible on the tissue surface (Fig. 5d). Consistently, we also detected increased expression of CCNH and γ-H2AX in candidalysin challenged mice compared to control group by IHC assay (Fig. 5d, e). The upregulation of γ-H2AX expression in WT and candidalysin treated mice demonstrated the presence of induced DSBs, through activation of CCNH expression, during mucosal infection.

## Discussion

Candidalysin plays a vital role in fungal infection. It can induce cell damage, calcium influx, membrane destabilization via membrane intercalation, permeabilization and pore formation to trigger epithelial immunity via c-Fos and MKP1 in the MAPK signaling pathway[18,19], including the release of IL-1α, IL-1β, IL-6, neutrophil recruitment, etc.[18,50–53]. But the direct human target of candidalysin remains unknown, which requires an overall understanding of the interactome between candidalysin and host. In this study, we performed a high-throughput enhanced yeast two-hybrid (HT-eY2H) assay, which provided stringent criteria to identify true protein interactions and generate high-quality interactome networks. The global screening and characterization of the interactome between Ece1 peptides and 13,761 human ORFs revealed candidalysin-specific biological function and pathways. This interactome could be used as a complete resource for exploration of the fungus–host interaction. Our results demonstrated CCNH as the direct target of candidalysin. CCNH, the highly conserved cyclin, can activate the cyclin-dependent kinase (CDK)-activating kinase (CAK) complex[54]. It is part of the transcription factor IIH multiprotein complex, which is required for RNA polymerase II

transcription and nucleotide excision repair[55]. Candidalysin can interact directly with CCNH to suppress DDR by activating the CAK complex, leading to severe host damage.

Previous work has shown that candidalysin damages the cell membrane to promote infection. However, the mechanism remains unclear. Russell, Schaefer *et al.* showed that candidalysin used a unique pore forming mechanism by creating a loop structure to insert into the membrane, leading to the membrane damage of human cells[56]. It would be interesting to investigate whether the CL mutation (G4W) or inhibitors that stop candidalysin from forming the polymers could induce the DNA damage in the future. Notably, far less attention has been given to the role of the other seven Ece1 peptides. In our study, we have also identified the host interactors of these Ece1 peptides. There are many overlapping or intersecting functions between the Ece1 peptides. For example, the members of the leukocyte immunoglobulin-like receptor (LIR) family, including *LILRA1*, *LILRA2*, *LILRA3*, *LILRB1*, *LILRB2*, *LILRB3*, *LILRB4* and *LILRB5*, can all interact with Ece1-II and Ece1-V peptides. These genes gather in some immune-related pathways like the immune response-inhibiting cell surface receptor signaling pathway, the B cell receptor signaling pathway or the immune response-inhibiting signal transaction. This was consistent with increasing evidence which have uncovered the mechanism of candidalysin in host immune responses, like *C. albicans* macrophage interaction[16,57]. Besides, we found that several host interactors of Ece1-I, Ece1-II, Ece1-IV, and Ece1-V peptides were enriched in keratinization pathways, including the keratin associated protein 10 family. Fungal infection was reported to cause keratinization of the skin and even symptoms such as onychomycosis[58]. In addition, dopamine and adenosine 3'5'-monophosphate-regulated phospho-protein (DARPP-32), located in the signal transduction pathway and integrating multiple intracellular signal transductions, plays a vital role in dopamine neurotransmission[59]. Interestingly, our bioinformatics analysis showed that host interactors were enriched in DARPP-32 events at high levels, and other interactors, including *RHOA* and *PPP2CA*, prompting us to explore the possibility of non-neurological functions of DARPP-32, especially its role in fungal infections.

Consistent with mycotoxins which can induce DNA damage in host, such as aflatoxins, ochratoxin A and zearalenone[60,61], candidalysin, was demonstrated in our study to induce DSBs and micronucleus formation. DNA damage typically trigger a downstream DDR system to maintain its overall integrity that involves nucleotide excision repair (NER), base excision repair (BER), DSB repair, and interstrand crosslink repair[46,60]. More interestingly, our study demonstrated that candidalysin can suppress DDR by simultaneous regulation of CCNH expression and CAK activity. Finally, in the oropharyngeal candidiasis animal experiments, we found that a single bolus dose of candidalysin administered orally could drive genotoxic damage to the same extent as fungal colonization that leads to extended candidalysin release shown by the micronucleus test. This might be due to the intricate complexity of host microenvironment and the regulation of other redundant pathways related to DNA damage in our interactome results, such as regulation of DNA repair and regulation of response to DNA damage stimulus.

Invasive fungal infections have become a high-risk factor of increasing morbidity and mortality in cancer patients. For example, *C. albicans* was closely associated with oral cancer which involved the induction of epithelial membrane damage, DNA damage and stimulation of inflammatory signaling pathways[23,62,63]. In addition, epidemiological studies have shown a significant overrepresentation of *C. albicans* in cases of colorectal cancer (CRC) or adenoma[64–66]. Increasing evidence has demonstrated that CCNH expression is upregulated with development of several types of cancers, such as gastrointestinal stromal tumors (GIST), breast cancer, esophageal squamous cell carcinoma and brain tumors[67–70]. Therefore, more functional insights into the mechanism of candidalysin-induced

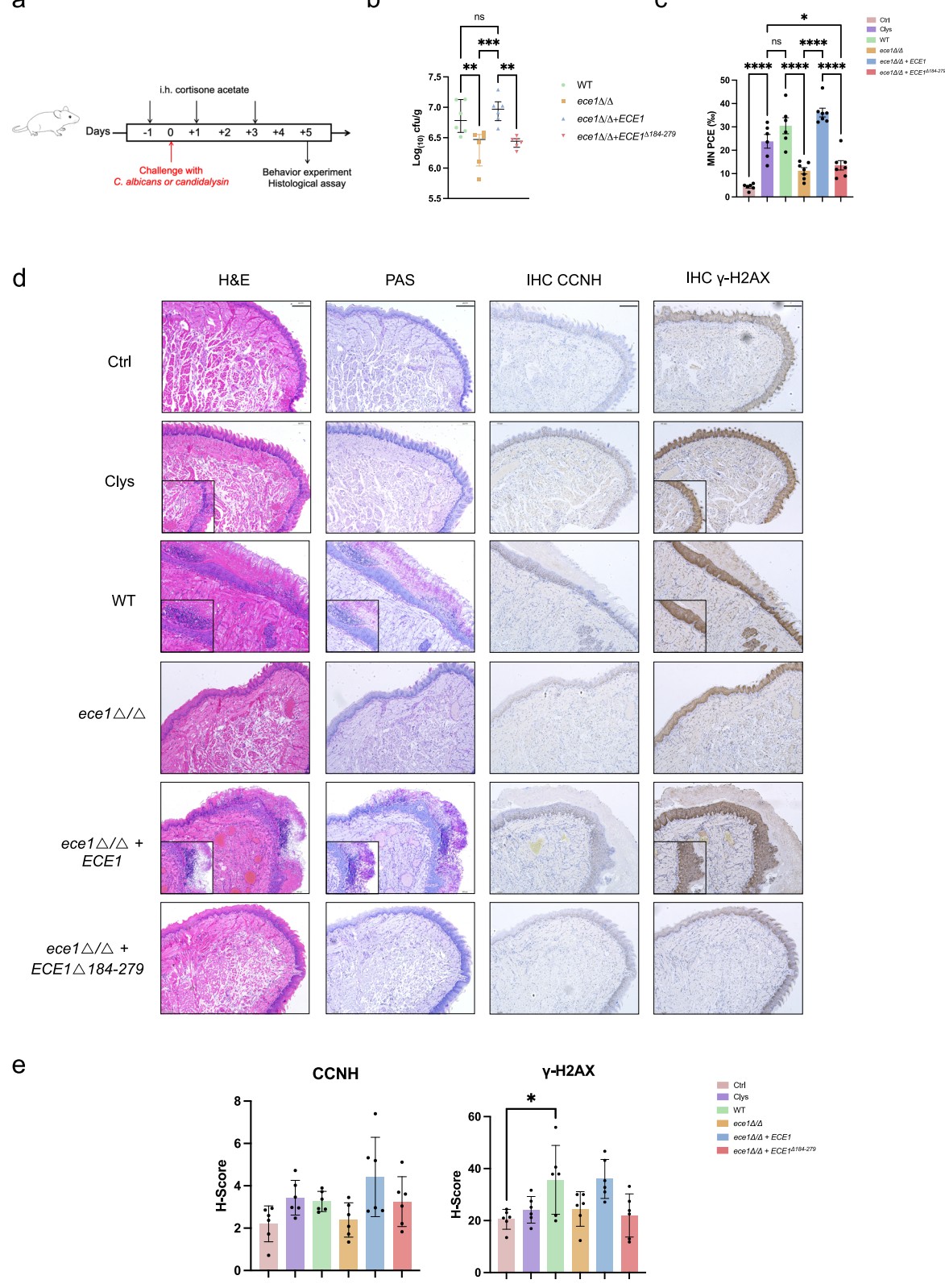

CCNH mediated fungal infection might elucidate its role in tumor progression.

Together, our study not only provide the human interactors of Ece1-peptides including candidalysin which may serve as potential therapeutic targets, but also make it possible in the future to analyze the genome-wide interactome between the CANDIDA ORFeome[71] and the Human ORFeome at the species level.

## Methods

### Ethics statement

All animal experiments are performed in compliance with the Regulations for the Care and Use of Laboratory Animals issued by the Ministry of Science and Technology of the People's Republic of China, which stipulates the ethical use of animals. The protocol has been approved by the Institutional Animal Care and Use Committee (IACUC)

**Fig. 5 | The CCNH and γ-H2AX expression was upregulated by candidalysin in vivo. a** As depicted in the schematics, on the day before infection (Day 0), mice (*n* = 6) were sedated by intraperitoneal injection of 75 mg/kg sodium pentobarbital. After anesthetization (Day 1), mice were infected with WT, *ece1Δ/Δ*, *ece1Δ/Δ+ECE1* and *ece1Δ/Δ+ECE1$^{Δ184-279}$ C. albicans*, or treated with 70 μM of candidalysin or PBS (Ctrl). After 5 days (Day 5), mice were euthanized. The tongue was excised and divided into two halves longitudinally for fungal load quantification and immunohistochemistry assay. Food-burial seeking tests were performed on the Day 0 and Day 5. **b** The colony forming units (CFUs) per gram of tongue tissue from mice colonized with WT, *ece1Δ/Δ*, *ece1Δ/Δ+ECE1*, and *ece1Δ/Δ+ECE1$^{Δ184-279}$* strains of *C. albicans*, and mice challenged with candidalysin or treated with PBS (Ctrl) (*n* = 6 mice). Data was shown as median with interquartile range. The *P* value was determined by one-way ANOVA with Tukey's post-hoc tests. *P* (WT vs. *ece1Δ/Δ*) = 0.0082; *P* (*ece1Δ/Δ vs. ece1Δ/Δ +ECE1*) = 0.0009; *P* (*ece1Δ/Δ+ECE1 vs. ece1Δ/Δ+ECE1$^{Δ184-279}$*) = 0.0041. **c** The ratio of

micronucleus PCE transformation (MN PCE‰) in peripheral blood of mice post infection at Day 5 (*n* = 6 mice). Data were shown as mean ± SEM. The *P* value was determined by one-way ANOVA with Tukey's post-hoc tests. *$P$ = 0.0243; ****$P$ < 0.0001. **d** Histopathology and immunohistochemistry assay of tongue tissues. Pathology and fungal infection in tongue tissues were assessed by H&E and PAS staining, respectively. The expression of CCNH and γ-H2AX was assessed in situ by immunohistochemistry assay (IHC). Scale bar, 200 μm. Images are representative of at least three individual experiments each performed in triplicate. Images are from the same tissues. **e** Quantification of CCNH and γ-H2AX expression in immunohistochemistry assay (*n* = 6 mice). The Histochemistry scores (H-Score) were calculated by the following formula: H-Score ($\sum$ (pi×i) = (percentage of weak intensity cells × 1) + (percentage of moderate intensity cells × 2) + (percentage of strong intensity cells ×3). Data were shown as mean ± SD. The *P* value was determined by one-way ANOVA with Tukey's post-hoc tests. *$P$ = 0.0286.

at Shanghai Institute of Immunity and Infection, Chinese Academy of Sciences (Permit Number: A2020071).

## Materials and resources

Most reagents used in this study (e.g. antibodies, chemicals, media, etc.) were purchased from reputable commercial vendors. As such, these reagents have passed a quality control screening process, and this information was provided to us upon purchase. Key resources used in this study are listed in Supplementary Table 1.

## Strain and plasmid construction

The strains used in the experiments are listed in the Supplementary Table 1. Plasmids are listed in Supplementary Table 2. All Ece1 related *C. albicans* strains used in this study were generous gifts from Dr. Bernhard Hube and Dr. David L. Moyes.

## Mouse strains

Female BALB/c mice aged 6-8 weeks were chosen and used in this study. Mice weighted 18-20 g were obtained from Shanghai Lingchang Biotechnology Co., LTD and randomly divided into five per cage. The same cage gives the same treatment. All the procedures were conducted in compliance with a protocol approved by the Institutional Animal Care and Use Committee at Institute Pasteur of Shanghai, CAS.

## The murine model of oropharyngeal candidiasis

As described in the previous murine oropharyngeal candidiasis model[18], on days −1, +1 and +3 post-infection, we took the average by weighing the mice, and injected the mice with 0.2 mL of cortisone acetate (225 mg/kg dissolved in PBS containing 0.05% (v/v) Tween 80) subcutaneously in the dorsum of the neck.

On the day before infection (Day 0), mice (*n* = 6) were sedated by intraperitoneal injection of 75 mg/kg sodium pentobarbital. After anesthetized, for *C. albicans* infected group, a sterile cotton swab soaked in 10$^7$ CFU/mL of *C. albicans* culture in PBS was placed on the tongue for 75 min, and for candidalysin treated group, a sterile cotton swab soaked in 70 μM candidalysin solution was placed on the tongue for 75 minutes. For the control group (*n* = 6), a swab soaked in PBS was used. After Day 5, mice were euthanized, the tongue was excised and collected for CFU counting, histopathology and immunohistochemistry analysis. The results of immunohistochemistry were quantified by histochemistry scores.

## Cell isolation

Bone marrow cells were removed from the femurs and tibias of BALB/c mice and processed with FBS. Prior to be enumerated under the microscope, the cells were air-dried naturally, fixed with methanol and stained with Giemsa solution (Sigma, 1.09203). Counts of at least 4,000 polychromatic erythrocytes (PCE) per piece were required to calculate the micronucleus PCE rate.

## Ece1 peptides

Peptides were synthesized and purchased from Shanghai Top-Peptide Biotechnology Co. Ltd. (China).

## Cell lines

The CHO-K1 (ATCC® SCSP-507) cell line were selected for the cell cycle analysis assay in this study. The FaDu (ATCC® TCHu132) and A549 (ATCC® CCL-185) cell lines were selected for the candidalysin damage assay used in this study. The CHO-K1 cell line was grown in F12K medium with 10% FBS, 1% Penicillin-Streptomycin (Pen-strep, Gibco, # 15140122) and 1% GlutMax (Gibco, # 35050061). CHO-K1, FaDu cells, and HEK-293T cells were kindly provided by Stem Cell Bank, Chinese Academy of Sciences, and have been tested error-free by STR. The FaDu cell line, A549 cell line and HEK-293T cell line were grown in DMEM medium with 10% FBS (Fetal bovine serum, Gibco, #10099-141) and 1% Penicillin-Streptomycin (Pen-strep, Gibco, # 15140122). Cells were maintained at 37 °C incubator with 5% CO$_2$.

## High-throughput enhanced yeast two-hybrid assay

The screening of potential host gene targets was carried out using our previous protocol with appropriate modification[26]. Briefly, this system is based on the reconstitution of a functional transcription factor for activation of HIS3 reporter in a cycloheximide (CHX)-sensitive manner through the interaction between two proteins: one fused to the DNA-binding domain (DB-X) and the other fused to the activation domain (AD-Y). The entry clone encoding candidalysin or other Ece1 peptides was generated by Gateway cloning and confirmed by variant identity through barcoding and Sanger sequencing. The entry clones were transferred by Gateway LR reactions into the HT-eY2H vector pDEST-AD expressing the yeast Gal4 activation domain fusion proteins (AD-Y). The AD-Y plasmids were then transformed into the haploid yeast Y8930 (*MATα*), with the genotype *leu2–3,112 trp1−901 his3Δ200 ura3−52 gal4Δ gal80Δ GAL2::ADE2 GAL1::HIS3@LYS2 GAL7::lacZ@MET2cyh2$^R$*, and selected on synthetic complete (SC) agar medium without leucine (SC-Leu) to generate Y2H bait strains. The prey strains expressing DB-X were transformed into the yeast Y8800, mating type *MATα*, with similar genotype as Y8930, and selected on SC medium without tryptophan (SC-Trp). The cells were then mated and spotted onto confirmative, selective and control agar plates including SC-Leu-Trp, SC-Leu-Trp-His and SC-Leu-His+CHX, respectively. PPIs were considered positive for activation of *GAL1::HIS3* (*i.e.*, growth to overcome 3AT inhibition) in a CHX-sensitive manner.

## Bioinformatics analysis

Statistical analysis was performed using R 4.0.2 and Python 3.7.6 with UpSetR[72], corrplot[73], pandas 1.0.1[74], and numpy 1.18.1[75]. The Fisher exact test was used for comparisons of overlaps of interacted gene between two peptides.

Gene set annotation and enrichment analysis were performed using Metascape[76] on the KEGG, GO, and Protein Atlas[77]. Firstly, pathway and process enrichment analysis has been carried out using default parameters with the following ontology sources: KEGG Functional Sets, GO Biological Processes, GO Cellular Components and GO Molecular Functions. The terms with the best $P$-values from each of the 20 clusters were visualized as a network using Cytoscape[78], where terms with a similarity > 0.3 were connected by edges. Secondly, protein–protein interaction enrichment analysis was carried out with the following databases: BioGrid[79], InWeb_IM[80], OmniPath[81]. The MCODE[82] inferred networks were also generated with Cytoscape.

## Cell cycle assay

For cell-cycle analysis, cells were collected and fixed overnight in 70% ethanol at −20 °C, washed and resuspended in 100 µg/mL RNase A Reagent for 30 min at 37 °C, then treated with 15 µg/mL propidium iodide (Elabscience, E-CK-A351) at 4 °C for 30 min in dark condition. After staining, cell-cycle analyses were carried out by flow cytometry. Data were analyzed with the FlowJo software (Tree Star). At least $1 \times 10^4$ cells were acquired per sample, and every experiment was repeated at least three times.

## LDH cytotoxicity assay

Cells were deposited into 96-well plates and incubated at 37 °C with 5% $CO_2$. Prior to the cell damage assay, cells were starved in serum-free medium overnight. Cytotoxicity LDH Assay Kit-WST (Dojindo, #CK12) was applied to determine the lactate dehydrogenase (LDH) activity released from cells that were damaged by candidalysin for 2 h or 4 h according to the manufacturer's instructions.

## Quantitative real-time PCR

To determine the gene expression patterns of FaDu and A549 cells, candidalysin-treated cells were washed with pre-warmed PBS and collected after addition of RNAlater (Beyotime, # R0118-100ml) before RNA isolation with a RNeasy Mini Kit (Qiagen, #74106). After determination of the purity and concentration of RNA by Nanodrop, the cDNA was obtained by a PrimeScript™ RT reagent Kit (Perfect Real Time) (Takara, #RR037A). Real-time PCR was then performed with 2x SYBR Green qPCR Master Mix (Bimake, # B21203) in a Biorad CFX96 thermocycler (CFX96 Touch Real-Time PCR Detection System). Primers used are as follows: ERCC1-internal-F (5'-AGTGGCCAAGCCCTTATTCC-3') and ERCC1-internal-R (5'-TGGCATATTCGGCGTAGGTC-3'), ERCC4-internal-F (5'-CAACGGCGGAGTTGTTTGAG-3') and ERCC4-internal-R (5'-TTGGCATTTGCAGCATTCCC −3'), ERCC5-internal-F (5'-GCAGC-CAGCGAAATAGAAGC-3') and ERCC5-internal-R (5'-TGCGAATCTGAAG-CACTGGT-3'), 53BP1-internal-F (5'-TCACAGAAAGTCCTCGTGCC-3') and 53BP1-internal-R (5'-CCGGTGTTGTCTCCACTCTC-3'), β-actin-internal-F (5'-CCTCGCCTTTGCCGATCC-3') and β-actin-internal-R (5'-GGAATCCT TCTGACCCATGC-3'), CCNH-internal-F (5'-GGAAGAATGGACTGATGAC GA-3') and CCNH-internal-R (5'-ACGTTTGATATGCTTCCTACTTCTC-3'), IL-1α-internal-F (5'-AGTGCTGCTGAAGGAGATGCCTG-3') and IL-1α-internal-R (5'-CCCTGCCAAGCACACCCAGTAG-3'), IL-6-internal-F (5'-CTCCTTCTCCACAAGCGCC-3') and IL-6-internal-R (5'-TGTGGGGCG GCTACATCTTTG-3'), RHOA-internal-F (5'-AGCCTGTGGAAAGA-CATGCTT-3') and RHOA-internal-R (5'- TCAAACACTGTGGGCACA-TAC-3').

## Western blot

Cells were lysed with cell lysis buffer (Beyotime, #P0013) on ice for 30 min. The supernatant of cell lysate was then collected in a table-top refrigerated centrifuge at 4 °C and 14,000 × $g$ for 10 min. Protein concentration was determined by Pierce™ BCA Protein Assay Kit (Thermo Scientific™, #23227). Antibodies against the following proteins were used for Western blotting analysis: CCNH antibody (CST,Rabbit Source, Cat#2927, 1:1000 dilution), gama-H2AX antibody (CST, Rabbit Source, Cat#97185, 1:1000 dilution), GAPDH (D16H11) XP® Rabbit mAb (CST, Rabbit Source, Cat#5174 S, 1:1000 dilution), Anti-CDK2 (phospho T160) + CDK1 (phospho T161) antibody (Abcam, Rabbit Source, Cat#EPR17621, 1:1000 dilution), Anti-rabbit IgG, HRP-linked Antibody (CST, Goat Source, Cat#7074, 1:5000 dilution), DYKDDDDK Tag (D6W5B) Rabbit mAb (CST, Rabbit Source, Cat#14793, 1:1000 dilution), GFP (4B10) Mouse mAb (CST, Mouse Source, Cat #2955, 1:1000 dilution), Anti-mouse IgG,HRP-linked antibody (CST, Goat Source, Cat#7076, 1:5000 dilution). Membranes were treated with SuperSignal™ West Pico PLUS Chemiluminescent Substrate (Thermo Scientific™, #34577) prior to visualization in the Chemiluminescence Imaging Analysis System (BG-gdsAUTO720, Baygene). GAPDH was used as the loading control.

## Micronucleus test (MNT)

About $3 \times 10^5$ FaDu cells were seeded into six-well culture plate for 24 h and exposed to either 10 µM candidalysin or 10 µM cisplatinum (micronuclei inductor) or their binary mixtures for 4 h. Cells were washed with fresh PBS and resuspended with fresh MEM medium with 10% FBS and 1% PS. The cells were further cultured for an extra 75 h. After washing with PBS, the treated cells were collected in 15 ml tube and fixed in methanol and acetic acid (3:1, v/v) overnight. The cultures were dropped onto glass slides, which were then stained with Giemsa solution for 5 min, washed with distilled water and dried at room temperature. For each treatment, 2000 cells were counted under ×400 magnification using a light microscope. The micronuclei frequency (MF) was thus calculated by (MN/2000) ×100.

## Comet assay

Alkaline comet analysis was performed with some modifications to evaluate DNA damage[83]. Briefly, following the termination of the treatment, $1.5 \times 10^4$ cells were mixed with 180 µl 0.5% low-melting-point agarose. The mixture was spread on pre-coated slides with normal agarose (1.5% in PBS) and cooled on ice to solidify. The slides were immersed in pre-cooled lysis solution (2.5 M sodium chloride, 100 mM EDTA, 10 mM Tris, 1% Triton-X 100, and 10% DMSO, pH 10) at 4 °C for 1 h, and placed in a gel electrophoresis apparatus containing electrophoresis buffer (300 mM NaOH and 1 mM EDTA) for 20 min to unwind the DNA, followed by electrophoresis for 20 min at 25 V / 300 mA on ice. Slides were neutralized (0.4 M Tris-HCl, pH 7.5) for 15 min and stained with 4′,6-diamidino-2-phenylindole (DAPI, Beyotime, #C1002). The comet images, taken by fluorescence microscopy (Olympus), were analyzed using CASP 1.2.3b1.

## Molecular docking

The structure of the eight Ece1 peptides were predicted using AlphaFold[84]. Molecular docking of candidalysin to CCNH was performed with the HPepDock server[48] (http://huanglab.phys.hust.edu.cn/hpepdock/). The structure of CCNH was fetched from Protein Data Bank[85] (PDB ID: 1KXU[47]), while the docking pose of candidalysin was sampled by the server based on the input sequence. Note that only the first 30 residues are selected for modeling due to restrictions by the webserver, so it would be worthy to evaluate the structural validity of the docking pose of candidalysin. Verification of modeled structures were performed using SAVES v6.0 (https://saves.mbi.ucla.edu/). Then the interaction surface analysis was done using Discovery Studio 2016. And the MM/GBSA study for energetical analysis of peptide binding was performed using the HawkDock server[86] (http://cadd.zju.edu.cn/hawkdock/). The binding free energy was calculated for the docking pose and further decomposed to each residue. More detailed information of molecular docking and analysis can be found in Supplementary Texts.

## Immunofluorescence assay

Cells were cultured on glass coverslips (#WHB-24-CS) in a 24-well plate. After candidalysin treatment, cells were fixed with 4% paraformaldehyde for 10 min and washed with prewarmed PBS for 3 times. Then cells were treated with 0.2% Triton X-100 for 5 min and blocked with PBS containing 10% FBS and 1% BSA for 1 h at 37 °C before incubation with primary antibody at 4 °C overnight. Then the coverslips with treated cells were immersed in HBSS containing 2 μM FM1-43 on ice for 1 min to stain the plasma membrane. After staining with antifade mounting medium with DAPI (Beyotime, #P0131), the cells were imaged under the confocal microscope. The antibodies used in this study are as follows: CCNH antibody (CST, Rabbit Source, Cat#2927, 1:500 dilution), Anti-gamma H2A.X (phospho S139) antibody (Abcam, Mouse Source, Cat#ab303656, 1:500 dilution), Goat Anti-Mouse IgG H&L (Alexa Fluor® 647) (Abcam, Goat Source, Cat#ab150115,1:500 dilution), Goat Anti-Rabbit IgG H&L (Alexa Fluor® 488) (Abcam, Goat Source, Cat#ab150077, 1:500 dilution).

## Co-immunoprecipitation

The overexpression plasmids CCNH-3×FLAG and Ece1-II-EGFP, Ece1-III-EGFP, Ece1-IV-EGFP and Ece1-V-EGFP were co-transfected into $5×10^5$ HEK-293T cells by Lipofectamine™ 3000 (Thermo Scientific™, L3000001) respectively. The HEK-293T cells were lysed for 48 h in RIPA buffer (Beyotime, P0013B) and added with the Phosphatase inhibitor cocktail D (Beyotime, P1096) and Phenylmethanesulfonyl fluoride (Beyotime, ST505) for 30 minutes on ice. After microcentrifuge for 15 minutes at 4 °C, the supernatants were heated at 100 °C for 5 min as the input group. The remaining supernatants were divided into two equal group: the GFP group added with GFP (5G4) Mouse mAb (Magnetic Bead Conjugate) (Cell Signaling Technology, #67090) at 1:10 dilution, and the IgG group added with Mouse IgG (Magnetic Bead Conjugate) (Cell Signaling Technology, #5873) at 1:10 dilution. Both groups were incubated with rotation overnight at 4 °C. By using magnetic rack to pellet beads, the samples were washed and lysised for western blot assay. Primers used are as follows: Candidalysin-VN173-internal-F (5′- ACAAAGACGATGACGA-CAAGCTTTCCATCATCGGCATCATCAT-3′) and Candidalysin-VN173-internal-R (5′- ATGGTGGCGATGGATCTTCTAGACTTATTGCCC TTGAAGGCCT-3′), CCNH-VC155-internal-F (5′-TATGGCCATG-GAGGCCCGAATTCGGATGTACCA CAACTCCAGC-3′) and CCNH-VC155-internal-R (5′-ATTTTGCACGCCGGACGGGTACCCAGGCTCTC-CACCAGGTCGT-3′). The antibodies used in this study were as follows: Mouse IgG (Magnetic Bead Conjugate) (CST, Mouse Source, Cat#5873, 1:10 dilution), GFP (5G4) Mouse mAb (Magnetic Bead Conjugate) (CST, Mouse Source, Cat#67090, 1:10 dilution).

## Surface plasmon resonance (SPR) measurement on BIAcore

Recombinant Cyclin H (CCNH) protein with a C-HIS tag (Lot. ME12NO2769, 38.46 kDa) was obtained from Sino Biological Inc. and stored at −80 °C. In short, the cyclin H (Homo sapiens) gene was amplified by PCR and cloned into an expression vector which was transformed into E. coli followed by the selection and establishment of high expression transformants. The protein was purified using immobilized metal ion affinity chromatography and dissolved in PBS (pH 7.4). SPR measurement was performed with Biacore T200 instrument (GE), at 25 °C, with a flow rate of 30 μL/min. CM5 sensor chips, HBS-EP running buffer and the Amine Coupling Kit, including 1-ethyl-3(3-dimethylaminopropyl) carbodiimide hydrochloride (EDC), N-hydroxysuccinimide (NHS), 1.0 M ethanolamine-HCl (pH 8.5) were purchased from GE. The binding properties of candidalysin to CCNH were evaluated upon immobilization of CCNH onto CM5 chip surface in flow cell 2 (FC-2) using Biacore Amine Coupling Kit according to the manufacturer's instructions. CCNH was diluted to 20 μg/ml in acetate buffer (pH 4.0) in advance and immobilized on the CM5 Chip following the EDC/NHS protocol. The reached immobilized target level was 1200

RU. The reference flow cell (FC-1) was blank-immobilized following the templated procedure. BIAcore measurements were performed starting from a 1:1 serial dilution of candidalysin following the templated multicycle analysis (association time 120 s, dissociation time 150 s, flow 30 μL/min). The SPR measurement for Ece1-II, Ece1-IV, Ece1-V and human cathelicidin (Solarbio, CLP0454) to CCNH were performed with Biacore 8 K instrument (GE) by PBST running buffer, and the other conditions were the same.

## Buried food-seeking test for the assessment of olfactory detection in mice

Buried food-seeking test was performed on the Day 0 before infection and on the Day 5. Foods were buried completely by litter in the upper left corner of the cage box (16 × 30 cm), and the mice began the exploration from the diagonal position of the cage box. The total time spent for locating and retrieving foods (latency to retrieve food) by each mouse was recorded (more than 10 min was recorded as a failed search). And then the video of each mouse retrieving the food was analyzed using VisuTrack Animal behavior analysis system (XinRuan Co. Ltd, Shanghai) for plotting the track and heatmap.

## Bimolecular fluorescence complementation (BiFC) assay

High-fidelity DNA polymerase (gene-optimal, 60070E) and the primers were used to amplify the target DNA of CCNH and candidalysin. The target DNA fragments were recovered and recombined with linearized pBiFC-VC155 (Vector), double cleaved by EcoRI (Thermo Scientific™, FD0583) and KpnI (Thermo Scientific™, FD0504), and pBiFC-VN173 (Vector), double cleaved by HindIII (Thermo Scientific™, FD0584) and XbaI (Thermo Scientific™, FD0503), respectively. Constructed pBiFC-VC155-CCNH and pBiFC-VN173-candidalysin were transformed into E. coli DH5α receptor cells (TIANGEN, CB101). The plasmids pBiFC-VC155-CCNH and pBiFC-VN173-candidalysin were co-transfected into $3 × 10^5$ HEK293T cells by Lipofectamine™ 3000 (Thermo Scientific™, L3000001). The pBiFC-VC155 (Vector) and pBiFC-VN173 (Vector) plasmid were co-transfected into the control group. Transfected cells were incubated at 37 °C for 12 h and then switched to 4 °C for 2 h. The cells were imaged under the fluorescence microscope.

## Calcein/PI cell viability assay

CHO-K1 cells were deposited into 96-well plates and incubated at 37 °C with 5% $CO_2$. Before the viability assay, cells were starved in serum-free F12K medium overnight. Calcein/PI Cell Viability/Cytotoxicity Assay Kit (DOJINDO, C542) was applied to determine the cell viability that were treated by candidalysin for 4 h.

## Co-culture of *C. albicans* and host cells

The wild type (SC5314), *ece1Δ/Δ*, *ece1Δ/Δ+ECE1* and *ece1Δ/Δ+ECE1^{Δ184-279}* *C. albicans* cells were individually inoculated into YPD broth and grown overnight at 30 °C and were harvested and washed three times with PBS, then counted using a hemocytometer. FaDu cells ($3 × 10^5$) were cultured in 24 mm Transwell® cell culture plates (Corning, 3450), and starved in serum-free DMEM medium overnight before co-culture. The *C. albicans* and FaDu cells were co-cultured in DMEM medium with 10% FBS, 1% Pen-strep, at 37 °C with 5% $CO_2$.

## RNA-seq analysis

FaDu cells ($3 × 10^5$) were cultured in 6-well plates (Thermo Scientific™, 140675), and starved in serum-free DMEM medium overnight before treatment. Ece1 peptides were dissolved in DMEM medium at a concentration of 10 μM, treated with FaDu cells for 4 h at 37 °C with 5% $CO_2$. The treated cells were washed with PBS and collected after addition of Trizol (TIANGEN, DP424).

Total RNA (~100 mg) was extracted from FaDu cells followed by library preparation according to Illumina standard instruction (VAHTS Universal V6 RNA-seq Library Prep Kit for Illumina®). Agilent 4200

bioanalyzer was employed to evaluate the concentration and size distribution of cDNA library before sequencing with an Illumina novaseq6000. The raw reads were filtered by Seqtk before mapping to genome using Hisat2 (version 2.0.4). The fragments of genes were counted using StringTie (v1.3.3b) followed by TMM (trimmed mean of M values) normalization. Significant differential expressed genes (DEGs) were identified as those with a False Discovery Rate (FDR) value above the threshold ($Q < 0.05$) and fold-change >2 using edgeR software.

### CCNH antibody blocking assay

Prior to stimulation with candidalysin, the FaDu cells were pre-incubated for 1 h with Cyclin H Antibody (Cell Signaling Technology, #2927) at 1:500 dilution or 1:1000 dilution. After 1 h, the blocked FaDu cells were stimulated with 10 μM candidalysin.

### Infection of fruit flies

The w1118 female flies at 3–5 days after eclosion were infected with four different *C. albicans* strains. Flies were anaesthetize on the pad under a light flow of $CO_2$. Then, flies were inoculate by intra-thoracic injection of 50.6 nL of *C. albicans* solution (1000 CFU, diluted with PBS buffer, pH 7.4) with fine glass capillaries using a Nanoject II automatic nanoliter injector (*Shuo Yang, 2019*). The injected flies grew at 29 °C, 60% humidity under a normal light/dark cycle and supply flies with fresh food daily.

### Testing drosophila olfaction with a Y-maze assay

To evaluate the chemosensory responses, a Y-maze Assay (*Mégane M. Simonnet, 2014*) was used at 6 dpi. The 40 μl acetic acid was added on one filter paper (~6 mm diameter), and 40 μl of distilled water was added on the second filter paper. An amount of 10 ~ 15 cold anesthetized flies were loaded into the start vial. The assembled Y-mazes were put in varying directions into a climate chamber at 23–26 °C and 50–70% humidity under far-red light for 24 h. The resulting olfactory index and acetic acid (AA) avoidance index were calculated using the following formulas: olfactory index = (number in the odor tube − number in the solvent tube)/total number of loaded flies; AA avoidance index = number in the odor tube/ total number of loaded flies. $N = 4$, representing a total of 81–143 flies.

### Statistical analyses

The Y2H screening outcomes were processed with ImageJ version 1.53 and then the data and gene lists were analyzed using R 4.0.2 and Python 3.7.6 with UpSetR[72], corrplot[73], pandas 1.0.1[74], and numpy 1.18.1[75]. Gene set annotation and enrichment analysis were performed using Metascape[76] and the results were shown with Cytoscape version 3.9.1. Gephi 0.9.2 was used for the Ece1 peptide-host protein interaction diagram.

Statistical tests were conducted with GraphPad Prism software 7.0 to compare experimental conditions and determine significance. The testing methods were selected based on appropriate assumptions with respect to type of question, null hypothesis, normality of the distributions and similarity of the variance. $P$ values < 0.05 were considered statistically significant. Test specifications, with the metrics and ranges displayed in the plots, were reported Figure legends or within the corresponding Method section.

### Reporting summary

Further information on research design is available in the Nature Portfolio Reporting Summary linked to this article.

## Data availability

All data needed to evaluate the conclusions in the paper, including data associated with main figures and Supplementary Figs., are available within the article or in the Supplementary Information, Source Data and Supplementary Data. Any additional information required to reanalyze the data reported in this paper is available from the lead contact upon request. The mRNA-seq analysis raw data generated in this study have been deposited in the NCBI database under accession number code BioProjectID: PRJNA1051990 and will be made publicly available upon publication. Source data are provided with this paper.

## Code availability

All code required to reproduce this study can be found at https://github.com/MiGiNull/Ece1p-interactome and https://doi.org/10.5281/zenodo.10615413[87].

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

## Acknowledgements

This study is supported by grants from the MOST Key R&D Program of China (2022YFC2304703, 2020YFA0907200), the National Natural Science Foundation (32270202, 82030099, 32311530119, 32170195), the National Key R&D Program of China (2022YFD2101500), the Science and Technology Commission of Shanghai Municipality (22DZ2303000), Program of Shanghai Academic/Technology Research Leader (23XD1422300), the Shanghai "Belt and Road" Joint Laboratory Project (22490750200), Innovative research team of high-level local universities in Shanghai, and the Innovative research team of high-level local universities in Shanghai. The authors thank Dr. Song Yi for providing the generous strains and plasmids, Dr. Geromy G. Moore for offering an extensive review of this manuscript prior to submission, and Instrumental Analysis Center of Shanghai Jiao Tong University for BIAcore experimental platform. All the candidalysin associated *C. albicans* strains are generous gifts from Dr. Bernhard Hube and Dr. David L. Moyes.

## Author contributions

N.-N.L., C.-B.C, and H.W. conceived and designed the study; N.-N.L., T.-Y.Z., and Y.-Q.C. performed the data analysis and wrote the manuscript; J.-C.T., T.-Y.Z., Y.-Q.C., J.-A.Z., B.-W.L.,T.J., L.-Q.W., Y.-M.H., X.-L.W., R.-Z.Z., J.X., Y.L., Y.Z., L.-Y.Z., and X.-K.Z. conducted all of the experiments; J.C.T., T.-Y.Z., and Y.-Q.C. performed the statistical analysis of the data; W.-N.C., Y.-Q.C., J.-Y.Z., Q.-Q.W., and W.-X.X. performed the bioinformatics analysis; N.-N.L., C.-B.C., H.W., L.P., J.Z., R.-X.Z., T.-Y.Z., Y.-Q.C. and J.-C.T. discussed the experiments and results.

## Competing interests

The authors declare no competing interests.
