## [Peer Review File · Nature Communications]

Global fungal-host interactome mapping identifies host targets of candidalysinREVIEWER COMMENTS

Reviewer #1 (Remarks to the Author):

In this study, the authors present a comprehensive interactome dataset using an enhanced yeast-2 hybrid system to identify potential human cellular interactors with the Candida toxin protein, Ece1. This is an important piece of work – particularly as it relates to the other Ece1 peptides apart from candidalysin, as they have no currently known function. In general, the work has been carried out to a good standard. However, the authors do not take the work too far with these toxins (which would be of particular value), and there are some important concerns around the work:

Abstract line 29. Host target for Cly isn't elusive – it's the membrane. Better to rephrase to intracellular targets have not been extensively mapped or some such.

What do the authors mean by an avirulence factor (line 98)? This doesn't sound correct. Do they mean that it functions to promote immune responses as well as being a virulence factor? This is NOT an avirulence factor, as it relates to the host and host responses.

Line 103 – saying that the human target of Cly remains unclear is not correct – we know that it targets the phospholipases in the cell membrane as a pore-forming toxin – in the original Nature paper. Direct human protein targets, however, aren't well known – you need to re-phrase this sentence and other occurrences in the manuscript.

Line 152. Were the proteins that interacted with the different individual peptides or groups of peptides from similar functional groups or specific pathways?

How do the authors explain the similarities in interactions between Candidalysin and ECE1-II, given the high degree of sequence and structural difference in the two peptides? Similarly for other shared genes.

Line 201 – the authors state olfactory receptor activity was ranked 1st and then go on to draw parallels and conclusions based around a role for Cly in olfaction and neurodegeneration. This is a bit strong, given that many proteins will be shared between different pathways – what evidence do the authors have to suggest their findings are specific to olfaction pathways? These need to be highlighted, or the statement amended.

Line 212 the authors state "For the other Ece1 peptides...". Is this a mistake, given that they've previously stated an analysis across all 8 peptides.

Line 230: The authors state that 10uM of Cly blocks cells in G2/M. This is an interesting finding, but given the lytic/toxic effects of Cly. it would be good to ensure that the doses of Cly used are not toxic to the cells.

Line 232 – the sentence starting "Somatic cells are strictly...". Doesn't appear to make any sense.

For γ -H2AX formation, CCNH expression etc, these are interesting findings, but could conceivably be part of a generic damage or pore-forming toxin effect. It would be good to see data using a non-toxin cytolysin control (e.g. low dose triton or similar) and a different pore forming toxin, e.g. mellitin or haemolysin etc. to determine how specific this response is to Cly or whether it represents a generic response to membrane damage. Also, what concentration of Cly was required for this action? No information is given for Fig 4d etc.

The authors have nice BIAcore evidence to indicate Cly interaction with CCNH, but can they show direct evidence of Cly binding to CCNH (or co-localisation) in vitro on treatment of cells with Cly?

The findings around erythrocytes are interesting – however, given the lower (non-existent) ability of ECE1 null strains to invade local tissue or to infect productively, how do the authors know that the differences are due to the lack of Cly, rather than simply a lack of infection? How do the ece1 null infected mice compare to sham infected animals?

The data is not sufficiently discussed in the discussion – there is too much emphasis on what other studies have shown, rather than discussing the findings shown here in this context.

Reviewer #2 (Remarks to the Author):

Summary

The objective of this manuscript by Zhang, et al. is to identify the mammalian molecular target(s) and additional function of the peptide toxin candidalysin (CL) secreted by the opportunistic fungus *Candida albicans* (Ca). In addition, the authors also sought to identify potential interactions of non-CL Ece1 peptides, which have not been ascribed functions to date. A yeast 2-hybrid (Y2H) approach was used in which individual Ece1 peptides were fused to the GAL4 activation domain (bait) and Gibson cloning used to create human ORFeome fusions to the GAL4 DNA binding domain (prey) to identify potential interacting partners. Bioinformatic analyses revealed a striking amount of unique predicted mammalian interactions amongst individual Ece1 peptides and several that have common shared targets. Ontological analysis suggested that most Ece1 peptides play a role in keratinization, while Ece1-III (i.e., CL) has predicted roles in signaling olfactory receptors and chromosome organization/cyclin D events in the G1 phase. Given the latter predicted role of CL, the authors then focus on CCNH, a subunit of a CDK-activating kinase important for cell cycle control. Flow cytometry revealed dysregulated cell cycle upon CL treatment and immunohistochemistry showed H2AX phosphorylation (a vital step in DNA damage repair). Expression of DNA repair related genes were up-regulated during short exposure of CL but not during extended exposure. Thus, the authors hypothesized that CL induces DNA damage and by binding to CCNH can suppress the DNA repair process. Surface plasmon resonance of recombinant CCNH with CL revealed a dissociation constant of 17 nM, suggesting strong affinity between these two molecules. Lastly, a mouse model of oropharyngeal candidiasis was performed using WT, *ece1* Δ/Δ , and Δ CL strains where CCNH, H2AX, and micronucleate polychromatic erythrocyte induction were interpreted to be CL-dependent, as was fungal burden.

While this is a very potentially exciting body of work, especially regarding potential interactions with non-CL Ece1 peptides, there a number of technical and theoretical concerns throughout. Please find more detailed comments below.

Major comments

1. While I can appreciate the enormous effort that went into performing the Y2H screen with each Ece1 peptide, there exists a very high false-positive rate with such an approach. The gold standard to validate findings is to perform co-immunoprecipitation. The authors should do this with tagged CCNH and Ece1-II, -III, -IV, -V since these were found as interactors in the Y2H screen. Do these others peptides also showing binding by surface plasmon resonance?
2. In the seminal paper in which CL was described (PMID: 27027296), only Ece1-II, -III, -V, and -VI were found in culture supernatants by LC-MS, suggesting that only these peptides are released extracellularly. In fact, Ece1-III (CL) was found to the greatest extent, and should the others get secreted they are likely a minority. Given these, what is the relevance of looking at interactors of most of the Ece1 peptides?
3. There are additional concerns about concentration dependence. For example, the authors should show that WT Ca infection drives similar DNA damage repair expression profiles, H2AX induction as presented in Fig. 4 and that an *ece1* Δ/Δ fails to do so. See additional comment below.
4. The animal data is troubling for a few reasons. The first, is that changes in MN PCE%, H2AX, and CCNH are marginal, and at times the data is not quantitative. Moreover, the *ece1* Δ/Δ and Δ CL mutants fail to colonize the murine tongue (100-fold less), so it is impossible to make interpretations about the role of CL in driving DNA damage. With significantly less fungus present, it could be due to any number of fungal effectors or host responses. Performing experiments in point 3 above might help clarify this.
5. A scrambled peptide or similar length amphipathic peptide (e.g. antimicrobial peptide) would have been incredibly helpful to rule out non-specific interactions. Due to its capacity to interact with membranes, CL could be associated with mammalian targets due to indirect interactions. Using a peptide with similar amphipathicity may help further delineate true targets from relatively non-specific interactors, especially given that the Ece1 peptides are expressed at high levels. This would also be a useful control for Biacore experiments.

6. Fig 4D. There appears to be very few cells in the control panel (should at least be DAPI+), so comparison against CL treatment is unfair. Experiment should be repeated and quantified.
7. Fig. 4G. Knockdown experiment results are not very convincing. There appears to be only a very modest increase in H2AX with CL-treatment. Moreover, why is this experiment done in an A549 (lung) cell line as opposed to oral epithelial cells used throughout and relevant to the animal experiments?
8. Fig. 4J. Unsure what the horizontal line is depicting. The figure likely needs edited.
9. This reviewer is not convinced that protein-protein interactions between CL and CCNH are driving DNA damage. Given its capacity to permeabilize membranes and lyse cells, it is possible that these processes in general are driving DNA damage and subsequent responses. The effects may not depend on specific interactions. The authors should utilize fluorescence microscopy to show that CCNH and CL specifically interact in cells or in vivo using a similar V5-epitope tagged CL as described (PMID: 33471869).
10. More detail regarding strain construction is required. Were these mutants acquired from the PIs who originally described CL or were they made in house? The authors state the strains/primers are listed in Table S2 but I do not see them.
11. Were the peptides expressed with 'K' or 'KR' as the terminal sequence? While they both induce damage similarly, the terminal R is typically cleaved by the Kex1p protease and the majority of peptides produced by Ca are of the 'K' variety. How does this impact interactions or data interpretation? It should at least be discussed.
12. Recently, it was shown that CL can polymerize into loops and insert into membranes (PMID: 36173096). How does this potentially impact its interaction with CCNH? This should be discussed.

Minor comments

1. The word "Candidalysin" should be changed to "candidalysin" throughout to be consistent with current usage and nomenclature for bacterial toxins.
2. A figure (perhaps new 1B) showing the sequence of each Ece1 peptide would be useful.
3. Line 80: better to state "including systemic or mucosal candidiasis". Otherwise, you leave out vaginal and cutaneous candidiasis.
4. Line 98: The term "avirulence factor" is odd. Perhaps "immunoavoidance"?
5. Line 201: insert "(Fig. 3b)" before the period.
6. Line 204: reference required.
7. Line 278-286. I find the description here quite confusing. You may wish to reword for clarity.
8. Fig 4i: I assume that CL is highlighted in pink? More description in the legend needed.
9. Fig. 5e: The statistical markers are placed over the wrong strains.
10. Line 386: reference incorrect here.
11. Line 397-403. I suggest moving this to the Results.
12. Line 413-415: I'm not sure what the authors mean. I do not see NLRP3 in the list of potential interactors. Please clarify.
13. Line 505: tense incorrect
14. Line 527-533: This is written like a protocol instead of a description of the procedure performed.

Reviewer #3 (Remarks to the Author):

The paper entitled "Global fungal-host interactome mapping identifies a novel target of Candidalysin for fungal infection" the authors performed an HT-eY2H assay to map the global interactomes of Ece1 peptides, including Candidalysin, with a part of the human ORFeome. They identified Candidalysin peptide target proteins in the host, and functionally characterized Candidalysin's interaction with CCHN, and its mechanism of action of this peptide. The manuscript is well executed, presents novel and interesting data to understand the pathogenesis of *C. albicans* infection. Only, there are points that they need to improve, which I describe below.

In the abstract the authors need to describe that they identified targets for the other Ece1 peptides, and that they also found shared targets between the peptides.

In the introduction, please report the sizes in amino acids of the rest of the peptides, they only describe that of Candidalysin.

The authors describe that "In this study, a HT-eY2H assay was used to map the global interactomes of Ece1 peptides, including Candidalysin, with the human ORFeome", I think it is important that they mention that it is a part of the human ORFeome, since there are only 13761 human proteins.

In results, the authors need to make a figure where they show the amino acid sequences of the 8 peptides of the Eci1 protein (I-VIII). They could identify signatures or boxes conserved between them, or also analyze if there is a correlation between the type of amino acids (for example hydrophobic or hydrophilic, etc) of these Eci1 peptides, and the target proteins that were identified in the two hybrids. All of the above, to analyze if any common region can be identified between the Ece1 peptides, when they identified overlapping human interacting proteins.

Another experiment that could be carried out is BiFC in human cells between the CCHN protein and the Candidalysin peptide, to observe in vivo the reconstitution of GFP fluorescence during the interaction of CCHN and the peptide.

Regarding the mutant *ece1Δ/Δ+ECE1Δ184-279*, the authors have to describe which peptides the deletion would cover, I suppose amino acids 184 to 279, and I believe that this region should contain the peptide Ece1-III (Candidalysin), is this correct? Please describe this part in detail.

In the discussion section:

"Thus, studying Ece1 peptide-host protein-protein interactions (PPIs) was crucial for understanding the mechanisms of fungal infection and the host response, and potentially to develop new strategies for fungal disease treatment and prevention" however, the authors only describe in This section mainly about the interactors of Candidalysin, and the other peptides discuss very briefly; I think that in the results section, they describe in greater detail, what they lack in discussion, for example, which are the human interactors that are particular to each Ece1 peptide, and which ones are overlapping. Also, the possible molecular mechanisms that each Ece1 peptide generates in the human interactome, and its effect on the infection of the fungus.

In this sentence: "Interaction networks are especially important as proteins generally act not in isolation but in concert with other proteins. Such interactomes can thus reveal biological pathways and processes impacted by the viral proteome, allowing for the discovery of novel drug targets"; however, I believe they are referring to the fungal proteome, rather than the viral proteome.

In the sentence "In addition, we also discovered that RHOA has the potential to be a host interactor as shown by an associated increase in mRNA and protein expression (Fig. S2a and b)", the authors have to describe which peptide(s) of Ece1 was identified as a RHOA interactor.

This paragraph "Through interactions of these host proteins with Ece1-II and Ece1-V peptides, we observed the synergistic or antagonistic mechanisms among Ece1 peptides and Candidalysin in host immune-related functions", is not very clear, the authors could improve it, on What synergistic or antagonistic mechanisms are they referring to?

Finally, the font size of most of the figures is very small, even some diagrams and graphs, please make it as big as possible.

Reviewer #4 (Remarks to the Author):

Authors of the presented manuscript uncovered potential host targets for *Candida albicans'* cytolytic peptide toxin, Candidalysin. Authors performed rigorous genomic-based studies and evaluated the potentiality of human CCNH through ex vivo and in vivo experimental investigations. Additionally, binding affinity between Candidalysin and its postulated target was assessed through molecular docking approach. The manuscript is valuable in its field as it redeems publication

following the address of these suggestions and comments.

1. Authors performed homology modelling to obtain the 3D structure of CCNH and Candidalysin for molecular docking analysis. However, CCNH atomic structure is actually deposited within the protein Data Bank database (<https://www.rcsb.org/>) PDB ID: 1KXU.

2. Furthermore, authors should made findings for the constructed structures using several online validation tools such as ERRAT, Verify3D and Rampage (like Z-score and Ramachandran plots) to be available even within the supplementary materials.

3. Zoomed 3D image(s) for CCNH-Candidalysin binding interface should be provided to highlight the polar and hydrophobic interactions between the two protein structures. Additionally, analysis for such binding interaction should be performed estimating the bond's angle and/or distances.

4. Authors are advised to explore the comparative mm-GBSA binding energies and its contributing energy terms (polar, hydrophobic, solvation, ...) using utilizing FastDRH web server (<https://cadd.zju.edu.cn/fastdrh/>) for the CCNH-Candidalysin complex. This would highlight the nature and magnitude of binding the thing that would guide further studies of optimization.

5. Performing short time Molecular Dynamic simulation (e.g. 50 ns) is advised which in turn would highlight the thermodynamic stability of Candidalysin at the CCNH binding site as well as potential target conformational alterations which would guide future interaction analysis.

Response to reviewers' comments

Reviewer #1:

In this study, the authors present a comprehensive interactome dataset using an enhanced yeast-2 hybrid system to identify potential human cellular interactors with the Candida toxin protein, Ece1. This is an important piece of work – particularly as it relates to the other Ece1 peptides apart from candidalysin, as they have no currently known function. In general, the work has been carried out to a good standard. However, the authors do not take the work too far with these toxins (which would be of particular value), and there are some important concerns around the work:

Answer: We are grateful to the reviewer for the pertinent comments. The manuscript has now been carefully revised and additional data were added in the revised manuscript, and the major revisions were marked with red color. The point-by-point responses to the reviewers' comments are shown as follows.

Q1. Host target for Cly isn't elusive – it's the membrane. Better to rephrase to intracellular targets have not been extensively mapped or some such. (Abstract line 29)

Answer: We appreciate the reviewer's suggestion and have rephrased "host targets" to "intracellular targets" in the revised manuscript. **(Line 31 on page 2)**

Q2. What do the authors mean by an avirulence factor (line 98)? This doesn't sound correct. Do they mean that it functions to promote immune responses as well as being a virulence factor? This is NOT an avirulence factor, as it relates to the host and host responses.

Answer: We thank the reviewer for this critical suggestion. We agree with the reviewer's comment that "avirulence" may mislead the readers, and therefore we

rephrased it to “immunoavoidance”, also suggested by another reviewer. **(Line 87 on page 4)**

Q3. Line 103 – saying that the human target of ClyA remains unclear is not correct – we know that it targets the phospholipases in the cell membrane as a pore-forming toxin – in the original Nature paper. Direct human protein targets, however, aren't well known – you need to re-phrase this sentence and other occurrences in the manuscript.

Answer: We apologize for the lack of clarity and thank the reviewer for raising this crucial point. The sentence was rewritten as “However, the direct human protein targets of candidalysin and the related mechanisms in promoting systemic fungal infection have not been fully defined.” Moreover, we also described the known target of candidalysin, the phospholipases in the cell membrane, in the revised Discussion section. **(Lines 33 to 35, 92 to 93 & 351 on pages 2, 4 & 13)**

Q4. Were the proteins that interacted with the different individual peptides or groups of peptides from similar functional groups or specific pathways? (line 152)

Answer: We thank the reviewer for raising this crucial point. Indeed, proteins that interact with different individual peptides or groups of peptides could be enriched in various pathways, which were further clustered as different functional groups **(Fig. R1, Extended data 2, Enrichment Sheet)**. The representative 20 pathways and corresponding genes were shown in **Supplementary Fig. 2**. For example, the interactors of Ece1-I, II and III could be enriched in the pathway of Cyclin D associated events in G1, which involves the genes such as CCNH, PPP2CA and THPO. Ece1-I, II, IV and V are all associated with Herpes simplex virus 1 infection, enriching genes like ZNF90 and EIF2B2. The detailed information about interactors of different peptides were listed in Extended data 2 (Annotation sheet), related to Fig. 3.

Fig. R1 Upset plot showing 20 pathways and the corresponding interactors which could be enriched in specific or shared pathways.

Q5. *How do the authors explain the similarities in interactions between Candidalysin and ECE1-II, given the high degree of sequence and structural difference in the two peptides? Similarly for other shared genes.*

Answer: Thanks for the reviewer’s valuable consideration. Although the sequences of these 8 Ece1 peptides are different, there exists structural binding similarity for Ece1-II and Ece1-III. Specifically, following the analysis of structural prediction with AlphaFold and molecular docking, we found that Ece1-II and Ece1-III could bind to host proteins such as THPO and SPRED2 with similar hydrophobic interactions, as shown below (**Fig. R2**). The docking results via HPepDock revealed that both Ece1-III and Ece1-II bind within the hydrophobic groove (yellow) of the THPO protein (PDB id: 8G04) or SPRED2 protein (PDB id: 2JP2). Similarly, this interaction pattern could lead to the relatively large number of share genes between these peptides.

Fig. R2 The structural binding similarity of Ece1-II and Ece1-III with THPO and SPRED2. a, the docking results via HPeDdock revealed that both Ece1-III and Ece1-II bind within the hydrophobic groove (yellow) of the THPO protein (PDB id: 8G04); b, the binding patterns of Ece1-II and Ece1-III with SPRED2 protein (PDB id: 2JP2). Both Ece1-III and Ece1-II bind within the hydrophobic groove (yellow) of the SPRED2.

Q6. *The authors state olfactory receptor activity was ranked 1st and then go on to draw parallels and conclusions based around a role for Clys in olfaction and neurodegeneration. This is a bit strong, given that many proteins will be shared between different pathways – what evidence do the authors have to suggest their findings are specific to olfaction pathways? These need to be highlighted, or the statement amended. (line 201)*

Answer: We appreciate the reviewer’s constructive suggestion and acknowledge the concern regarding the strong statement linking Clys to olfaction and neurodegeneration. We have conducted two behavioral experiments specifically targeting olfactory function in mice (food-burial search test) and fruit flies (Olfactory Choice Test, Y-maze Assay) to explore the impact of either wild type or *ece1* mutant infection on the olfactory capabilities of experimental animals. The results revealed that mice infected

with candidalysin-null mutant spent more time trying to acquire food (longer latency to retrieve food), suggesting potential detrimental effects of Cly5 on host olfactory function (**Supplementary Fig. 3a & 3b**). Furthermore, the fruit flies infected with WT *Candida albicans* showed olfactory dysfunction compared to control or the *ece1Δ/Δ+ECEI^{Δ184-279}* infected group. Both the olfactory index and AA avoidance index were decreased in fruit flies affected by Cly5 knockout mutant, suggesting the compromised AA avoidance behavior. These results should provide reasonable evidence for a possible association between Cly5 and olfactory impairment. Following the reviewer's suggestion, we also amended our statement regarding to the influence on neurodegenerative diseases in the manuscript. These new figures and corresponding modifications are provided in the revised manuscript. (**Lines 186 to 195 on pages 7 to 8**)

Q7. The authors state “For the other Ecel peptides...”. Is this a mistake, given that they've previously stated an analysis across all 8 peptides. (line 212)

Answer: We apologize for this mistake and have deleted the phrase “For the other Ece1 peptides” to make it coherent. (**Line 196 on page 8**)

Q8. The authors state that 10uM of Cly5 blocks cells in G2/M. This is an interesting finding, but given the lytic/toxic effects of Cly5. it would be good to ensure that the doses of Cly5 used are not toxic to the cells. (line 230)

Answer: We appreciate the reviewer's suggestion and have evaluated the cytotoxicity of candidalysin in CHO-K1 cells (4hrs) via cell activity assay and cytotoxicity LDH assay (**Fig. 4a**). The results of cellular toxicity test showed that 10 μM of Cly5 was not able to induce toxic damage to host cells. The figures and corresponding main text have been adapted. (**Lines 221 to 223 on page 9**)

Q9. The sentence starting “Somatic cells are strictly...”. Doesn't appear to make any sense. (line 232)

Answer: Thanks for the reviewer's valuable consideration and we have deleted this sentence. (Line 225 on page 9)

Q10. For γ -H2AX formation, CCNH expression etc, these are interesting findings, but could conceivably be part of a generic damage or pore-forming toxin effect. It would be good to see data using a non-toxin cytolysin control (e.g. low dose triton or similar) and a different pore forming toxin, e.g. mellitin or haemolysin etc. to determine how specific this response is to Clys or whether it represents a generic response to membrane damage. Also, what concentration of Clys was required for this action? No information is given for Fig 4d etc.

Answer: We are grateful to the reviewer for pointing our attention to this important issue and this constructive suggestion. We have performed additional experiments to detect the expression levels of CCNH and γ -H2AX in FaDu cells following treatments of Clys, mellitin, hemolysin and triton-X100. Although these results indicated a relatively modest increase of CCNH expression by these toxin and non-toxin cytolysin, only Clys induced a significantly elevated expression of γ -H2AX, substantiating the specific role of Clys in inducing DNA damage (DSB). All these modifications are adapted in the revised manuscript. (Supplementary Fig. 5d)

In addition, the concentration of Clys in the fluorescence colocalization assay in Fig. 4d (new Fig. 4e) is 10 μ M. This detail information has been added in the figure legend and results.

Q11. The authors have nice BIAcore evidence to indicate Clys interaction with CCNH, but can they show direct evidence of Clys binding to CCNH (or co-localisation) in vitro on treatment of cells with Clys?

Answer: We thank the reviewer for raising this crucial point. We have performed co-immunoprecipitation assays to validate the interaction between intracellular Clys and CCNH. Plasmids encoding 3xFlag-tagged CCNH and Ece1-II to V peptides constructs

linked with EGFP were transiently transfected into HEK-293T cells. The co-immunoprecipitation results demonstrated that Clys binds to CCNH intracellularly (**Fig. 4i**). To strengthen this point, we performed the bimolecular fluorescence complementation (BiFC) assay by transfecting N-terminal EGFP-tagged Clys and C-terminal EGFP-tagged CCNH into HEK-293T cells. We observed the green fluorescence signal, confirming the interaction between Clys and CCNH proteins within the host cells (**Fig. 4j**). All these modifications are adapted in the revised manuscript. (**Lines 277 to 282 on page 11**)

Q12. The findings around erythrocytes are interesting – however, given the lower (non-existent) ability of ECE1 null strains to invade local tissue or to infect productively, how do the authors know that the differences are due to the lack of Clys, rather than simply a lack of infection? How do the ece1 null infected mice compare to sham infected animals?

Answer: We appreciate the reviewer for raising this crucial point. We have repeated the fungal infection experiment in mice, with the introduction of an additional Clys-infected group. The results of micronucleus tests showed a significant increase in the bone marrow micronucleus rate (MN PCE%) in mice infected with Clys compared to the control group (**Fig. 5c**). However, there was no statistically significant difference compared to mice infected with the WT strain, confirming the genotoxicity role of Clys during *Candida albicans* infection in mice. Furthermore, for the Ece1 null strain infected group, the MN PCE% was similar to that of sham infected group. These results indicate that the reduced MN PCE rates observed in mice infected with the *ece1Δ/Δ* and *ece1Δ/Δ+ECE1^{Δ184-279}* strains are possibly due to the absence of Clys rather than the decreased fungal load. All these modifications are adapted in the revised manuscript. (**Lines 326 to 331 on pages 12 to 13**)

Q13. The data is not sufficiently discussed in the discussion – there is too much emphasis on what other studies have shown, rather than discussing the findings shown here in this context.

Answer: We appreciate the reviewer for raising this crucial point. We have reorganized the discussion part in the revised manuscript as follows.

(1) Summary: The paragraph “Candidalysin plays a vital role in fungal infection...” and “In this study, ...”

(2) Discussion about candidalysin as a pore-forming toxin and discussion about interaction of host protein targets and other Ece1 peptides: “Previous work has shown that candidalysin damages the cell membrane to promote infection. However, how candidalysin does this remained unclear. Russell, Schaefer et al. showed that candidalysin used a unique pore forming mechanism by creating a loop structure to insert into the membrane, leading to the membrane damage of human cells. It would be interesting to investigate whether the CL mutation (G4W) or inhibitors that stop candidalysin from forming the polymers could induce the DNA damage in the future. Notably, far less attention has been given to the role of the other seven Ece1 peptides. In our study, we have also identified the host interactors of these Ece1 peptides. There are many overlapping or intersecting functions between the Ece1 peptides. For example, ...”

(3) Discussion about the role of candidalysin in DNA damage and tumorigenesis: The paragraph “Consistent with mycotoxins which can induce DNA damage in host, ...” and the paragraph “Invasive fungal infections have become a high-risk factor of increasing morbidity and mortality in cancer patients. For example, ...”

(4) Outlook of further study: “Together, our study not only provide the human interactors of Ece1-peptides including candidalysin which may serve as potential therapeutic targets, but also make it possible in the future to analyze the genome-wide interactome between the CANDIDA ORFeome and the Human ORFeome at the species level.” **(Lines 347 to 413 on pages 13 to 16)**

Reviewer #2:

Summary

*The objective of this manuscript by Zhang, et al. is to identify the mammalian molecular target(s) and additional function of the peptide toxin candidalysin (CL) secreted by the opportunistic fungus *Candida albicans* (Ca). In addition, the authors also sought to identify potential interactions of non-CL Ece1 peptides, which have not been ascribed functions to date. A yeast 2-hybrid (Y2H) approach was used in which individual Ece1 peptides were fused to the GAL4 activation domain (bait) and Gibson cloning used to create human ORFeome fusions to the GAL4 DNA binding domain (prey) to identify potential interacting partners. Bioinformatic analyses revealed a striking amount of unique predicted mammalian interactions amongst individual Ece1 peptides and several that have common shared targets. Ontological analysis suggested that most Ece1 peptides play a role in keratinization, while Ece1-III (i.e., CL) has predicted roles in signaling olfactory receptors and chromosome organization/cyclin D events in the G1 phase. Given the latter predicted role of CL, the authors then focus on CCNH, a subunit of a CDK-activating kinase important for cell cycle control. Flow cytometry revealed dysregulated cell cycle upon CL treatment and immunohistochemistry showed H2AX phosphorylation (a vital step in DNA damage repair). Expression of DNA repair related genes were up-regulated during short exposure of CL but not during extended exposure. Thus, the authors hypothesized that CL induces DNA damage and by binding to CCNH can suppress the DNA repair process. Surface plasmon resonance of recombinant CCNH with CL revealed a dissociation constant of 17 nM, suggesting strong affinity between these two molecules. Lastly, a mouse model of oropharyngeal candidiasis was performed using WT, *ece1* Δ/Δ , and Δ CL strains where CCNH, H2AX, and micronucleate polychromatic erythrocyte induction were interpreted to be CL-dependent, as was fungal burden.*

While this is a very potentially exciting body of work, especially regarding potential interactions with non-CL Ece1 peptides, there a number of technical and theoretical concerns throughout. Please find more detailed comments below.

Answer: We appreciate the reviewer's detailed and helpful comments. The manuscript has now been carefully revised and additional data were added in the revised manuscript, with the major revisions marked with red color. Please take a look at the following point-by-point responses addressing the reviewer's comment.

Major comments

***Q1:** While I can appreciate the enormous effort that went into performing the Y2H screen with each Ece1 peptide, there exists a very high false-positive rate with such an approach. The gold standard to validate findings is to perform co-immunoprecipitation. The authors should do this with tagged CCNH and Ece1-II, -III, -IV, -V since these were found as interactors in the Y2H screen. Do these other peptides also showing binding by surface plasmon resonance?*

Answer: We appreciate the reviewer for raising this crucial point. We have conducted co-immunoprecipitation assays to validate the direct interaction between intracellular Ece1 peptides and CCNH. Plasmids encoding 3xFlag-tagged CCNH and Ece1-II to V peptides constructs linked with EGFP were transiently transfected into HEK-293T cells. The co-immunoprecipitation results demonstrated the interaction between intracellular CCNH and Ece1 peptides (Ece1-II to V). (**Supplementary Fig. 5a**)

Furthermore, we validated the binding of Ece1-II to V peptides to CCNH through surface plasmon resonance assays (via BIAcore). The results revealed a strong binding affinity between these peptides and CCNH, as compared to the negative control antimicrobial peptide (Amp). This finding was consistent with our results from the Y2H screening. Among the peptides, the strongest binding observed is CCNH-Clys (Ece1-III). (**Supplementary Fig. 5g & h**)

Thus, both co-IP and BIAcore assays validated the Y2H screening results. All these modifications are adapted in the revised manuscript. (Lines 217 to 221 on page 9)

Fig. R3 The binding efficiency of Ece1-II, IV, V and Amp (antimicrobial peptide) to CCNH protein was evaluated by BIAcore.

Q2: *In the seminal paper in which CL was described (PMID: 27027296), only Ece1-II, -III, -V, and -VI were found in culture supernatants by LC-MS, suggesting that only these peptides are released extracellularly. In fact, Ece1-III (CL) was found to the greatest extent, and should the others get secreted they are likely a minority. Given these, what is the relevance of looking at interactors of most of the Ece1 peptides?*

Answer: We appreciate the reviewer for raising our attention to this point. In this seminal paper, the supernatant used for LC-MS detection of peptides was from fungal cell culture, but not co-cultures of fungi with cells. We hypothesized that the secretion profile of peptides might differ from that after co-culturing with host cells. We treated FaDu cells separately with Ece1 peptides and then conducted RNA-seq analysis on them to study the changes in cellular gene expression post-peptide treatment. The

results revealed the potential different functions of each peptide during fungal infection (**Supplementary Fig. 1**). That's why we performed the Y2H screening of the interactors of all the Ece1 peptides. All these modifications have been supplemented in the revised manuscript. (**Lines 103 to 112 on page 5**)

Q3: *There are additional concerns about concentration dependence. For example, the authors should show that WT Ca infection drives similar DNA damage repair expression profiles, H2AX induction as presented in Fig. 4 and that an *ece1Δ/Δ* fails to do so. See additional comment below.*

Answer: We thank the reviewer for this crucial point. Firstly, we determined the concentration of Clys by infecting cells with different concentrations of Clys to examine DNA damage. Our findings revealed a concentration-dependent effect of Clys on DNA damage, with 10μM concentration as the most significant upregulation of γ-H2AX. (**Supplementary Fig. 5e**)

Secondly, we performed the infection assay by co-culturing cells with the *C. albicans* WT (SC5314) and *ece1Δ/Δ*. Then the expression levels of γ-H2AX and DNA damage repair associated genes (53BP1, ERCC1, ERCC2, ERCC4) were measured. We observed a significant activation of γ-H2AX expression in both WT and *ece1Δ/Δ+ECE1* strains, indicating DNA double-strand breaks caused by the strains. Consistently, we also found the upregulation of DNA damage repair associated genes (53BP1, ERCC1, ERCC2, ERCC4) after WT *C. albicans* infection, while the *ece1Δ/Δ* strain failed to do so (**Supplementary Fig. 5c**). All these modifications have been supplemented in the revised manuscript. (**Lines 256 to 257 on page 10**)

Q4: *The animal data is troubling for a few reasons. The first, is that changes in MN PCE%, H2AX, and CCNH are marginal, and at times the data is not quantitative. Moreover, the *ece1Δ/Δ* and *CLA* mutants fail to colonize the murine tongue (100-fold less), so it is impossible to make interpretations about the role of CL in driving DNA damage. With significantly less fungus present, it could be due to any number of fungal*

effectors or host responses. Performing experiments in point 3 above might help clarify this.

Answer: We thank the reviewer for this critical comment. We have repeated the fungal infection experiment in mice, with the introduction of an additional Clys-infected group. The results of micronucleus tests showed a significant increase in the bone marrow micronucleus rate (MN PCE‰) in mice infected with Clys compared to the control group (**Fig. 5c**). However, there was no statistically significant difference compared to mice infected with the WT strain, confirming the genotoxicity role of Clys during *Candida albicans* infection in mice. Furthermore, for the Ece1 null strain infected group, the MN PCE‰ was similar to that of sham infected group. These results indicate that the reduced MN PCE rates observed in mice infected with the *ece1Δ/Δ* and *ece1Δ/Δ+ECE1^{Δ184-279}* strains are possibly due to the absence of Clys rather than the decreased fungal load. Furthermore, the results of IHC assay also indicated the DNA damage in the tissue caused by Clys and strains capable of secreting Clys (WT and *ece1Δ/Δ+ECE1*) in mice, along with the relative upregulation of CCNH expression. All these data have been quantified (**Fig. 5e**). Thus, the aboving results indicated the role of Clys in driving DNA damage. All these modifications are adapted in the revised manuscript. (**Fig. 5**)

Q5: *A scrambled peptide or similar length amphipathic peptide (e.g. antimicrobial peptide) would have been incredibly helpful to rule out non-specific interactions. Due to its capacity to interact with membranes, CL could be associated with mammalian targets due to indirect interactions. Using a peptide with similar amphipathicity may help further delineate true targets from relatively non-specific interactors, especially given that the Ece1 peptides are expressed at high levels. This would also be a useful control for Biacore experiments.*

Answer: We appreciate the reviewer for this pertinent comment. We followed the reviewer's suggestion and used an antimicrobial peptide (Amp) as a negative control in the BIAcore experiment. The equilibrium dissociation constant of Clys-CCNH binding

is $K_D=1.720\times 10^{-8}$ M ($pK_D=7.76$) while that of Amp-CCNH binding is $K_D=6.26\times 10^{-5}$ M ($pK_D=4.20$). This result validated the relatively high affinity binding between Clys and CCNH. (**Fig. 4k and Supplementary Fig. 5g & h**)

Q6: *Fig 4D. There appears to be very few cells in the control panel (should at least be DAPI+), so comparison against CL treatment is unfair. Experiment should be repeated and quantified.*

Answer: We thank the reviewer for raising this issue. We have repeated the infection assay with an equivalent number of cells and performed the co-localization immunofluorescence analysis of CCNH and γ -H2AX between the Clys-treated and control groups within the field of view using ImageJ software. After infection with Clys, we observed the increased co-localization of CCNH and γ -H2AX intracellularly. All these modifications are adapted in the revised manuscript. (**Fig. 4e**)

Q7: *Fig. 4G. Knockdown experiment results are not very convincing. There appears to be only a very modest increase in H2AX with CL-treatment. Moreover, why is this experiment done in an A549 (lung) cell line as opposed to oral epithelial cells used throughout and relevant to the animal experiments?*

Answer: We apologize for this confusion. We actually have tried several times to knockout CCNH in FaDu cells (oral epithelial cells used throughout) by using the CRISPR/Cas9 system. However, we observed a significantly low survival rate which prevented the selection of the positive depletion mutant cells (**Fig. R4**). We hypothesized that this might be due to the essential role of CCNH in FaDu cells, the depletion of which led to cell death. Additionally, as a supplement, we have conducted a CCNH blocking experiment by incubating cells with CCNH antibodies before Clys infection *in vitro* to examine the expression of γ -H2AX. As shown in Supplementary Fig. 4f, we discovered that inhibition of CCNH led to a suppressed expression of γ -H2AX within the cells, revealing the role of CCNH in mediating Clys-induced

inhibition of DNA damage repair (**Supplementary Fig. 5f**). All these modifications are adapted in the revised manuscript. (**Lines 273 to 176 on page 11**)

Fig. R4 Representative images of cell morphology during CCNH knockout by CRISPR/Cas9.

Q8: Fig. 4J. Unsure what the horizontal line is depicting. The figure likely needs edited.

Answer: We have updated the new figure in our revised manuscript. (**Fig. 4m**)

Q9: *This reviewer is not convinced that protein-protein interactions between CL and CCNH are driving DNA damage. Given its capacity to permeabilize membranes and lyse cells, it is possible that these processes in general are driving DNA damage and subsequent responses. The effects may not depend on specific interactions. The authors should utilize fluorescence microscopy to show that CCNH and CL specifically interact in cells or in vivo using a similar V5-epitope tagged CL as described (PMID: 33471869).*

Answer: We are grateful to the reviewer for this constructive suggestion. According to the reviewer's suggestion and the cited reference (PMID: 33471869), we have performed the bimolecular fluorescence complementation (BiFC) assay by transfecting

N-terminal EGFP-tagged Clys and C-terminal EGFP-tagged CCNH into HEK-293T cells. We observed the green fluorescence signal, confirming the interaction between Clys and CCNH proteins within the host cells (**Fig. 4j**). These results showed that CCNH and Clys specifically interact in cell. All these modifications are adapted in the revised manuscript. (**Lines 277 to 282 on page 11**)

Q10: *More detail regarding strain construction is required. Were these mutants acquired from the PIs who originally described CL or were they made in house? The authors state the strains/primers are listed in Table S2 but I do not see them.*

Answer: We apologize for this lack of clarity. These mutant strains were all acquired from Dr. Bernhard Hube and Dr. David L. Moyes. We have added the details of strains and primers used in this study into **Supplementary Table 1 & 2** respectively.

Q11: *Were the peptides expressed with 'K' or 'KR' as the terminal sequence? While they both induce damage similarly, the terminal R is typically cleaved by the Kex1p protease and the majority of peptides produced by Ca are of the 'K' variety. How does this impact interactions or data interpretation? It should at least be discussed.*

Answer: We thank the reviewer for pointing out our attention to this lack of clarity. The Ece1-III peptide used in this study was terminated with the 'K' residue, which is the dominant form in *C. albicans* infection (PMID: 27027296). It was previously reported that the terminal R is typically cleaved by the Kex1p protease and the majority of peptides produced by *C. albicans* are of the 'K' variety. While they both induce damage similarly, we used the Ece1-III peptide with 'K' in this study to largely mimic *in vivo* situation. We have discussed this in our revised manuscript.

Q12: *Recently, it was shown that CL can polymerize into loops and insert into membranes (PMID: 36173096). How does this potentially impact its interaction with CCNH? This should be discussed.*

Answer: We very much appreciate the reviewer's suggestion. Previous work has shown that candidalysin damages the cell membrane to promote infection. However, how candidalysin does this remained unclear. Russell, Schaefer *et al.* showed that candidalysin used a unique pore forming mechanism by creating a loop structure to insert into the membrane to form a pore, leading to the membrane damage of human cells (PMID: 36173096). This type of pore formation has not been observed before. It would be interesting to investigate whether the CL mutation (G4W) or inhibitors that stop candidalysin from forming the polymers could induce the DNA damage in the future. We have updated the discussion of the main text accordingly. **(Lines 365 to 370 on page 14)**

Minor comments

Q1: *The word "Candidalysin" should be changed to "candidalysin" throughout to be consistent with current usage and nomenclature for bacterial toxins.*

Answer: We thank the reviewer and have updated the word "candidalysin" throughout the manuscript.

Q2: *A figure (perhaps new 1B) showing the sequence of each Ece1 peptide would be useful.*

Answer: We appreciate the reviewer for this crucial suggestion. We have added a figure **(Fig. 1a)** showing the sequence of each Ece1 peptide.

Q3: *Line 80: better to state "including systemic or mucosal candidiasis". Otherwise, you leave out vaginal and cutaneous candidiasis.*

Answer: We thank the reviewer have rephrased it to "including systemic or mucosal candidiasis". **(Lines 70 to 71 on page 4)**

Q4: *Line 98: The term "avirulence factor" is odd. Perhaps "immunoavoidance"?*

Answer: We thank the reviewer and the term “avirulence factor” has been changed into “immunoavoidance”. **(Line 87 on page 4)**

Q5: Line 201: insert “(Fig. 3b)” before the period.

Answer: We thank the reviewer for raising this point and “(Fig. 3b)” has been added. **(Line 192 on page 8)**

Q6: Line 204: reference required.

Answer: We thank the reviewer for reminding us. The reference has been added into this statement. **(Line 186 on page 7)**

Q7: Line 278-286. I find the description here quite confusing. You may wish to reword for clarity.

Answer: We apologize for our confusing description, and we have reworded this part as follows. “Importantly, a long-term exposure of candidalysin leads to sustained higher expression of γ -H2AX even during the recovery period, an indicator of ineffectiveness of the DDR. For comparison, following the short-term exposure, the expression level of γ -H2AX was gradually restored to control level (Fig. 4g). Intriguingly, we observed a dose-dependent inhibition of CCNH expression by candidalysin after the 24h treatment, while γ -H2AX level remained elevated. This observation suggests that a long-term exposure of candidalysin may cause irreversible effects on DDR.” **(Lines 257 to 264 on page 10)**

Q8: Fig 4i: I assume that CL is highlighted in pink? More description in the legend needed.

Answer: We thank the reviewer’s comment. Detailed description has been added in the legend. **(Fig. 4i)**

Q9: Fig. 5e: The statistical markers are placed over the wrong strains.

Answer: We apologize for this mistake and have made a new **Fig. 5c**.

Q10: *Line 386: reference incorrect here.*

Answer: We thank the reviewer for reminding us. The reference was inserted here by mistake and we have removed it. **(Line 366 on page 14)**

Q11: *Line 397-403. I suggest moving this to the Results.*

Answer: We thank the reviewer for this nice suggestion. We have moved this part to the results. **(Lines 232 to 237 on page 9)**

Q12: *Line 413-415: I'm not sure what the authors mean. I do not see NLRP3 in the list of potential interactors. Please clarify.*

Answer: We apologize for making the reviewer confused. Actually, NLRP3 is not in the list of potential interactors. We have reworded this part as follows. "This was consistent with increasing evidence which have uncovered the mechanism of candidalysin in host immune responses, like *C. albicans* macrophage interaction." **(Lines 379 to 381 on page 14)**

Q13: *Line 505: tense incorrect.*

Answer: We thank the reviewer for point this. The tense here has been corrected. The word "inject" has been changed into "injected". **(Line 440 on page 18)**

Q14: *Line 527-533: This is written like a protocol instead of a description of the procedure performed.*

Answer: We apologize for the writing in this section of method and we have rewritten this part as follows. "Bone marrow cells were removed from the femurs and tibias of BALB/c mice and processed with FBS. Prior to be enumerated under the microscope, the cells were air-dried naturally, fixed with methanol and stained with Giemsa solution

(Sigma, 1.09203). Counts of at least 4,000 polychromatic erythrocytes (PCE) per piece were required to calculate the micronucleus PCE rate.” **(Lines 453 to 457 on page 18)**

Reviewer #3 (Remarks to the Author):

The paper entitled “Global fungal-host interactome mapping identifies a novel target of Candidalysin for fungal infection” the authors performed an HT-eY2H assay to map the global interactomes of Ece1 peptides, including Candidalysin, with a part of the human ORFeome. They identified Candidalysin peptide target proteins in the host, and functionally characterized Candidalysin's interaction with CCHN, and its mechanism of action of this peptide. The manuscript is well executed, presents novel and interesting data to understand the pathogenesis of C. albicans infection. Only, there are points that they need to improve, which I describe below.

Answer: We thank the reviewer for the constructive suggestions. We have now revised our manuscript by supplementation with additional experiments and analyses accordingly. Please see the following point-by-point responses addressing the reviewer’s comments. All revisions are highlighted with red color in the revised manuscript.

Q1: In the abstract the authors need to describe that they identified targets for the other Ece1 peptides, and that they also found shared targets between the peptides.

Answer: We thank the reviewer for raising this clarification point. We added in the abstract that we identified targets for the other Ece1 peptides, and that we also found shared targets between the peptides. **(Line 33 on page 2)**

Q2: In the introduction, please report the sizes in amino acids of the rest of the peptides, they only describe that of Candidalysin.

Answer: We thank the reviewer for this suggestion and we have added the new **Fig.1a** showing the sequence of the 8 Ece1 peptides in the manuscript. Furthermore, the sequence and the sizes in amino acids peptide is shown in the table below.

Peptide	Sequence	Size	Mass (Da)
---------	----------	------	--------------

Ece1-I	MKFSKIACATVFALSSQAIIHHAPEFNMKR	31	3449.13
Ece1-II	DVAPAAPAAPADQAPTVPAPQEFNTAITKR	30	3016.32
Ece1-III	SIIGIIMGILGNIPQVIQIIMSIVKAFKGNKR	32	3310.12
Ece1-IV	EDIDSVVAGIADMPFVVRAVDTAMTSVASTKR	33	3465.94
Ece1-V	DGANDDVANAVVRLPEIVARVATGVQQSIENAKR	34	3576.95
Ece1-VI	DGVDPVGLNLVANAPRLISNVFDGVSETVQQAKR	34	3580.01
Ece1-VII	DGLEDFLDELLQRLPQLITRSAESALKDSQPVKR	34	3882.39
Ece1-VIII	DAGSVALSNLIKKSIETVGIENAAQIVSERDISSLIEEYFGKA	43	4567.13

Q3: *The authors describe that "In this study, a HT-eY2H assay was used to map the global interactomes of Ece1 peptides, including Candidalysin, with the human ORFeome", I think it is important that they mention that it is a part of the human ORFeome, since there are only 13761 human proteins.*

Answer: We thank the reviewer for point this. We have reworded the statement about human ORFeome as "a part of the human ORFeome (n=13,761)". **(Line 95 on page 4)**

Q4: *In results, the authors need to make a figure where they show the amino acid sequences of the 8 peptides of the Ece1 protein (I-VIII). They could identify signatures or boxes conserved between them, or also analyze if there is a correlation between the type of amino acids (for example hydrophobic or hydrophilic, etc) of these Ece1 peptides, and the target proteins that were identified in the two hybrids. All of the above, to analyze if any common region can be identified between the Ece1 peptides, when they identified overlapping human interacting proteins.*

Answer: We thank the reviewer for giving this significant suggestion. We have added a figure **(Fig. 1a)** showing the amino acid sequences of 8 Ece1 peptides (I-VIII). We also conducted the conserved sequence analysis using the online tool of multiple sequence alignment (*Clustal Omega*, <https://www.ebi.ac.uk/Tools/>) and we found low

homology or similarity between the Ece1 peptides as shown below (**Fig. R5**). We hypothesized that the similarity in interactors of these peptides may be associated with the similar intermolecular interaction between them.

Fig. R5 Conserved sequence analysis using the online tool of multiple sequence alignment. a, multiple sequence alignment of Ece1-I to VIII peptides. The number right of the sequence represents the length of each peptide; b, the 8 Ece1 peptides were clustered according to the homology score. The results of multiple sequence alignment and homology analysis showed low similarity between the Ece1 peptides.

Q5: Another experiment that could be carried out is BiFC in human cells between the CCHN protein and the Candidalysin peptide, to observe in vivo the reconstitution of GFP fluorescence during the interaction of CCHN and the peptide.

Answer: We appreciate the reviewer for this nice suggestion and have performed the bimolecular fluorescence complementation (BiFC) assay by transfecting N-terminal EGFP-tagged ClyS and C-terminal EGFP-tagged CCNH into HEK-293T cells. We observed the green fluorescence signal, confirming the interaction between ClyS and CCNH proteins within the host cells (**Fig. 4j**). (**Lines 277 to 282 on page 11**)

Q6: *Regarding the mutant $ece1\Delta/\Delta+ECE1\Delta184-279$, the authors have to describe which peptides the deletion would cover, I suppose amino acids 184 to 279, and I believe that this region should contain the peptide Ece1-III (Candidalysin), is this correct? Please describe this part in detail.*

Answer: We apologize for this confusion. The ECE1 184-279 was the nucleotide sequence, which corresponds to amino acids 62-93 in the peptide sequence. This detail has been added in the revised manuscript. **(Line 324 on page 12)**

Q7: *In the discussion section:*

"Thus, studying Ece1 peptide-host protein-protein interactions (PPIs) was crucial for understanding the mechanisms of fungal infection and the host response, and potentially to develop new strategies for fungal disease treatment and prevention" however, the authors only describe in This section mainly about the interactors of Candidalysin, and the other peptides discuss very briefly; I think that in the results section, they describe in greater detail, what they lack in discussion, for example, which are the human interactors that are particular to each Ece1 peptide, and which ones are overlapping. Also, the possible molecular mechanisms that each Ece1 peptide generates in the human interactome, and its effect on the infection of the fungus.

Answer: We appreciate the reviewer for raising this crucial point. We have reorganized the discussion part in the revised manuscript as follows.

(1) Summary: The paragraph "Candidalysin plays a vital role in fungal infection..." and "In this study, ..."

(2) Discussion about candidalysin as a pore-forming toxin and discussion about interaction of host protein targets and other Ece1 peptides: "Previous work has shown that candidalysin damages the cell membrane to promote infection. However, how candidalysin does this remained unclear. Russell, Schaefer et al. showed that candidalysin used a unique pore forming mechanism by creating a loop structure to

insert into the membrane, leading to the membrane damage of human cells. It would be interesting to investigate whether the CL mutation (G4W) or inhibitors that stop candidalysin from forming the polymers could induce the DNA damage in the future. Notably, far less attention has been given to the role of the other seven Ece1 peptides. In our study, we have also identified the host interactors of these Ece1 peptides. There are many overlapping or intersecting functions between the Ece1 peptides. For example, ...”

(3) Discussion about the role of candidalysin in DNA damage and tumorigenesis: The paragraph “Consistent with mycotoxins which can induce DNA damage in host, ...” and the paragraph “Invasive fungal infections have become a high-risk factor of increasing morbidity and mortality in cancer patients. For example, ...”

(4) Outlook of further study: “Together, our study not only provide the human interactors of Ece1-peptides including candidalysin which may serve as potential therapeutic targets, but also make it possible in the future to analyze the genome-wide interactome between the CANDIDA ORFeome and the Human ORFeome at the species level.” **(Lines 347 to 413 on pages 13 to 16)**

Q8: In this sentence: “Interaction networks are especially important as proteins generally act not in isolation but in concert with other proteins. Such interactomes can thus reveal biological pathways and processes impacted by the viral proteome, allowing for the discovery of novel drug targets”; however, I believe they are referring to the fungal proteome, rather than the viral proteome.

Answer: We apologize for this mistake and we deleted it. **(Line 358 on page 14)**

Q9: In the sentence “In addition, we also discovered that RHOA has the potential to be a host interactor as shown by an associated increase in mRNA and protein expression (Fig. S2a and b)”, the authors have to describe which peptide(s) of Ece1 was identified as a RHOA interactor.

Answer: We appreciate the reviewer's suggestion and have listed the Ece1 peptides that were identified as RHOA interactors. They are Ece1-I, II, VI, VIII. **(Lines 242 to 243 on page 9)**

Q10: This paragraph "Through interactions of these host proteins with Ece1-II and Ece1-V peptides, we observed the synergistic or antagonistic mechanisms among Ece1 peptides and Candidalysin in host immune-related functions", is not very clear, the authors could improve it, on What synergistic or antagonistic mechanisms are they referring to?

Answer: We apologize for confusing the reviewer in this paragraph. To avoid ambiguity, we have revised this statement as follows. "This was consistent with increasing evidence which have uncovered the mechanism of candidalysin in host immune responses, like C. albicans macrophage interaction." **(Lines 379 to 381 on page 14)**

Q11: Finally, the font size of most of the figures is very small, even some diagrams and graphs, please make it as big as possible.

Answer: We thank the reviewer for this crucial suggestion and we have made the font size of figures as big as possible.

Reviewer #4:

Authors of the presented manuscript uncovered potential host targets for Candida albicans' cytolytic peptide toxin, Candidalysin. Authors performed rigorous genomic-based studies and evaluated the potentiality of human CCNH through ex vivo and in vivo experimental investigations. Additionally, binding affinity between Candidalysin and its postulated target was assessed through molecular docking approach. The manuscript is valuable in its field as it redeems publication following the address of these suggestions and comments.

Answer: We are grateful to the reviewer for the insightful comments and constructive suggestions. The manuscript has now been carefully revised and additional data were added in the revised manuscript following the reviewer's suggestion. Please see the following point-by-point responses addressing the reviewer's comment. All revisions in the main text are highlighted in red.

Q1: Authors performed homology modelling to obtain the 3D structure of CCNH and Candidalysin for molecular docking analysis. However, CCNH atomic structure is actually deposited within the protein Data Bank database (<https://www.rcsb.org/>) PDB ID: 1KXU.

Answer: We thank the reviewer for this crucial suggestion. Actually, the final docking pose was acquired by adapting chain A of 1KXU as the structure of receptor. We apologize for the ambiguity in the Methods section and we have clarified the docking procedure in the **Method of Molecular docking. (Lines 562 to 575 on pages 22 to 23)**

Q2: Furthermore, authors should made findings for the constructed structures using several online validation tools such as ERRAT, Verify3D and Rampage (like Z-score and Ramachandran plots) to be available even within the supplementary materials.

Answer: We thank the reviewer for this critical suggestion and we have validated the constructed structures using the online validation tool ERRAT of SAVES server to generate Ramachandran plot and ERRAT score.

(1) Ramachandran plot

The Ramachandran plot (**Supplementary Fig. 6a**) was plotted using SAVES v6.0 (<https://saves.mbi.ucla.edu/>). It could be observed that 91.3% of the residues locate in the most favored regions, above the recommended 90%. No residue locates in disallowed regions. The plot suggests the validity of backbone torsions.

(2) ERRAT Score

The ERRAT score is calculated using the SAVES v6.0 webserver. The score has a value of 68.1818, suggesting a moderate good standard of structure from the perspective of crystallography.

We have added a section of **verification of modeled structures** in the Method of Molecular Docking and **Supplementary Texts** for detailed information. Based on previous analyses, it could be inferred that sampled docking pose by HPepDock server is reasonable in structure.

Q3: Zoomed 3D image(s) for CCNH-Candidalysin binding interface should be provided to highlight the polar and hydrophobic interactions between the two protein structures. Additionally, analysis for such binding interaction should be performed estimating the bond's angle and/or distances.

Answer: We thank the reviewer for the advice. We have added zoomed 3D image(s) for CCNH-candidalysin binding interface in both the section of Methods of the main text and the Supplementary Texts. The zoomed 3D as well as 2D interaction plot are provided and the key interactions are highlighted. (**Supplementary Fig. 6b**)

Specifically, the molecular docking of candidalysin to CCNH was performed with the HPepDock server (<http://huanglab.phys.hust.edu.cn/hpepdock/>). The structure of CCNH was fetched from Protein Data Bank (PDB ID: 1KXU), while the docking pose of candidalysin was sampled by the server based on the input sequence. Note that the only first 30 residues are selected for modeling due to restrictions by the webserver. Therefore, it would be worthy to evaluate the structural validity of the docking pose of candidalysin.

To gain deep insights into the CCNH-candidalysin binding, we first analyzed the interaction surface. The analysis was done using Discovery Studio 2016 and 3D as well as 2D interaction plots (**Supplementary Fig. 6b**). It could be observed that the binding is mostly triggered by hydrophobic interaction, but not a moderate H-bond between Trp11 on CCNH and Gly11 on candidalysin. The relative residues and distance of interactions are labeled. The types of interactions are listed in **Supplementary Fig. 6b**.

***Q4:** Authors are advised to explore the comparative mm-GBSA binding energies and its contributing energy terms (polar, hydrophobic, solvation, ...) using utilizing FastDRH web server (<https://cadd.zju.edu.cn/fastdrh/>) for the CCNH-Candidalysin complex. This would highlight the nature and magnitude of binding the thing that would guide further studies of optimization.*

Answer: We thank the reviewer for the suggestion. Unfortunately, we have tried the FastDRH web server and it could not be used to properly read our input. We have contacted the developers of the server but we haven't got any feedbacks. Therefore, we turned to another server, HawDock (<http://cadd.zju.edu.cn/hawkdock/>) for MM/GBSA calculations. This section has been added as the comparative MM/GBSA binding energies calculation in the **Supplementary Texts**.

The binding free energy was calculated for the docking pose and was further decomposed to each residue. It could be observed in the following table that the Top 5 residues contributing the binding is Ile5, Ile3, Ile13, Pro14 and Gly4, which was

consistent with our results that these residues dominate the protein-peptide binding via hydrophobic interactions. (**Supplementary Fig. 6b and Supplementary Table 4**)

Q5: Performing short time Molecular Dynamic simulation (e.g. 50 ns) is advised which in turn would highlight the thermodynamic stability of Candidalysin at the CCNH binding site as well as potential target conformational alterations which would guide future interaction analysis.

Answer: We thank the reviewer for the suggestion. We have performed MD simulations followed by conformation and distance analysis of the trajectories. This section has been added as short time Molecular Dynamic simulation analysis in both the main text and the **Supplementary Texts**.

The complex was solvated with a TIP3P box and neutralized with *tleap* in AMBER18 package. The system was minimized and then heated to 300K in NVT ensemble within 100 ps. After that, the system was equilibrated in turn in NVT and NPT ensemble, each for 100 ps. Finally, the system was simulated for 50 ns using NPT ensemble. The minimization and simulation were performed with the GPU-accelerated form of *pmemd* in AMBER18 package using the ff14SB forcefield.

The trajectory was processed with principal component analysis (PCA). The equilibrated conformation (Equil. Conf.) was extracted as the conformation around (-4.6, -0.5) in **Supplementary Fig. 6c**. It could be observed in **Supplementary Fig. 6d** that the equilibrated conformation of the peptide is almost the same as that of the docking pose. Instead, the C-terminal of the peptide forms two additional H-bonds with CCNH, causing the previous erect N-terminal fall down to the surface of CCNH. This is rational because in molecular docking, solvation effects could not be considered. Besides, we have monitored the distances representing hydrophobic interactions in the docking pose. It could be seen in **Supplementary Fig. 6e** that these distances are mostly stably maintained around those in the docking pose, suggesting the stability of the peptide-protein interactions.

REVIEWERS' COMMENTS

Reviewer #1 (Remarks to the Author):

Generally, this revised manuscript is much improved, and my main questions fully answered. I have only a couple of further points.

Firstly, in answer to the Q5, regarding similarity of peptide binding, the authors need to include this in the text to ensure that there is no confusion for the reader.

Regarding Q6, the authors have made great strides in answering the question and generally I am happy with and believe their data. However, it is notable that they have still not definitively answered the issue - it is still possible that the loss of olfaction that they observe was due to destruction of olfactory cells, rather than due to any interference with olfactory receptors. It is important that they note this in the text at least, given the fact that the peptides they are studying include a cytolysin.

Finally, although the new text is generally well written and cohesive, there are a few grammatical errors that should be corrected.

Reviewer #1 (Remarks on code availability):

I'm not a coding expert.

Reviewer #2 (Remarks to the Author):

The authors have been very responsive to disparate and comprehensive prior critiques raised by the reviewers. Aided by additional experimentation, they now provide stronger evidence for candidalysin-host target interactions in driving DNA damage. Specifically, confirmatory co-immunoprecipitation assays and BiaCore significantly add to the manuscript to support the conclusions. Additional non-candidalysin toxicity controls (e.g., mellitin) also demonstrate specificity. New animal experiments suggest that candidalysin activity and not necessarily fungal colonization drive DNA damage and that candidalysin interrupts olfaction. In vitro infections also support the animal studies. There are a few lingering questions/comments that could help improve the manuscript further.

1. It is surprising that a single bolus dose of candidalysin administered orally could drive DNA damage to the same extent as fungal colonization that leads to extended candidalysin release (fig 4c). It is difficult to explain why such damage would not have resolved by day 5 post-challenge. The authors should comment on this.
2. Fig 4a. Please change "infection" to "challenge" as mice cannot be infected with candidalysin.
3. The "Amp" control peptide used in fig. S4g and 4k should be defined as "human cathelicidin" in the text or legend.
4. The bifunctional fluorescence data shown in fig. 4j is interesting, but the interactions seem infrequent (or just a small sample size). In any case, the HEK293T cells should be counterstained for better visualization.

Reviewer #3 (Remarks to the Author):

The authors have answered all my observations.

Reviewer #4 (Remarks to the Author):

Authors adequately responded to all comments and suggestions

Response to reviewers' comments

Reviewer #1 (Remarks to the Author):

Generally, this revised manuscript is much improved, and my main questions fully answered. I have only a couple of further points.

We thank the reviewer for this informative feedback. We have revised our manuscript accordingly. Please see the following point-by-point responses addressing the reviewer's comments. All revisions to the main text were highlighted in red.

Q1. Firstly, in answer to the Q5, regarding similarity of peptide binding, the authors need to include this in the text to ensure that there is no confusion for the reader.

Answer: We appreciate the reviewer for this critical suggestion. We have included this similarity of peptide binding in the corresponding result part of the main text as follows: "The relatively large number of shared genes between these peptides may be a result of the similar molecular interaction patterns." **(Lines 159 to 160 on page 6)**

Q2. Regarding Q6, the authors have made great strides in answering the question and generally I am happy with and believe their data. However, it is notable that they have still not definitively answered the issue - it is still possible that the loss of olfaction that they observe was due to destruction of olfactory cells, rather than due to any interference with olfactory receptors. It is important that they note this in the text at least, given the fact that the peptides they are studying include a cytolysin.

Answer: We thank the reviewer for raising this crucial point. We agree with the reviewer's comments and have mentioned the possibility in the relevant result part. "However, we cannot exclude the possibility that the detrimental effects of candidalysin treatment on host olfactory function might also be due to the destruction of olfactory cells, considering the property of a cytolysin." **(Lines 197 to 200 on page 8)**

Q3. Finally, although the new text is generally well written and cohesive, there are a few grammatical errors that should be corrected.

Answer: We are grateful to the reviewer for pointing the errors. We have checked our manuscript throughout and corrected the grammatical errors. All revisions to the main text were highlighted in red. **(Line 109 on page 5, lines 218, 220, 221, 243 on page 9, line 308 on page 12, line 412 on page 15)**

Reviewer #1 (Remarks on code availability):

I'm not a coding expert.

Reviewer #2 (Remarks to the Author):

The authors have been very responsive to disparate and comprehensive prior critiques raised by the reviewers. Aided by additional experimentation, they now provide stronger evidence for candidalysin-host target interactions in driving DNA damage. Specifically, confirmatory co-immunoprecipitation assays and BIAcore significantly add to the manuscript to support the conclusions. Additional non-candidalysin toxicity controls (e.g., mellitin) also demonstrate specificity. New animal experiments suggest that candidalysin activity and not necessarily fungal colonization drive DNA damage and that candidalysin interrupts olfaction. In vitro infections also support the animal studies. There are a few lingering questions/comments that could help improve the manuscript further.

We are grateful to the reviewer for the generous comments and constructive suggestions. The manuscript has been carefully revised following the reviewer's suggestions. Please see the following point-by-point responses addressing the reviewer's comments. All revisions to the main text have been highlighted in red.

***Q1.** It is surprising that a single bolus dose of candidalysin administered orally could drive DNA damage to the same extent as fungal colonization that leads to extended candidalysin release (Fig. 4c). It is difficult to explain why such damage would not have resolved by day 5 post-challenge. The authors should comment on this.*

Answer: We appreciate the reviewer for raising our attention to this point. First, in the co-culturing cell line experiments (Fig. 4g), we observed that the DNA damage was resolved within 24 hours. Second, in the oropharyngeal candidiasis animal experiments, we found that a single bolus dose of candidalysin administered orally could drive DNA damage to the same extent as fungal colonization that leads to extended candidalysin release shown by the micronucleus test. Actually, the micronucleus test can assess the genotoxicity caused by candidalysin, including not only the DNA damage, but also the chromosomal breaks, etc. Furthermore, compared with the direct treatment of

candidalysin on host cells, the host-fungal interaction in the oropharyngeal candidiasis animals might encounter a range of intrinsic signals originating from both host immunity and fungal pathogenicity. There exists the intricate complexity of host-fungal interaction in the animal experiments. We speculate that the signaling pathways induced by *in vivo* genotoxic damage are highly complicated. In addition, the regulation of other redundant pathways related to DNA damage such as regulation of DNA repair and regulation of response to DNA damage stimulus in our interactome results (Fig. 3b & Extended Data 2), might also be affected by candidalysin treatment. Considering the complexity of animal experiments, it may be insufficient for mice to recover from the genotoxic damage caused by direct candidalysin peptide treatment within five days. Thus, these comprehensive factors could result in a comparable micronucleus rate (MN PCE‰) in the candidalysin-treated group with that of the WT strain colonization group. **(Lines 403 to 409 on page 15)**

Q2. Fig 5a. Please change “infection” to “challenge” as mice cannot be infected with candidalysin.

Answer: We thank the reviewer for this pertinent comment. We have replaced “infection” with “challenge” in **Fig. 5a** and the corresponding main text. **(Line 346 on page 13, and line 902 on page 41)**

Q3. The “Amp” control peptide used in Fig. S5g and 5h should be defined as “human cathelicidin” in the text or legend.

Answer: We are grateful to the reviewer for this suggestion. We have defined “Amp” as “human cathelicidin” in the corresponding main text and the figure legends of Fig. S5g and 5h. **(Line 667 on page 26, and line 879 on page 38)**

Q4. The bifunctional fluorescence data shown in Fig. 4j is interesting, but the interactions seem infrequent (or just a small sample size). In any case, the HEK293T cells should be counterstained for better visualization.

Answer: We appreciate the reviewer for raising this crucial point. We have followed the reviewer's suggestion and counterstained the HEK293T cells, which were shown in the new **Fig. 4j**.

Reviewer #3 (Remarks to the Author):

The authors have answered all my observations.

We appreciate the reviewer for this generous comment.

Reviewer #4 (Remarks to the Author):

Authors adequately responded to all comments and suggestions.

We are grateful for the reviewer for this generous comment.